# All-optical spatiotemporal mapping of ROS dynamics across mitochondrial microdomains in situ

Shon A. Koren[1], Nada Ahmed Selim[2], Lizbeth De la Rosa[1], Jacob Horn[1], M. Arsalan Farooqi[1], Alicia Y. Wei[1], Annika Müller-Eigner [3], Jacen Emerson[1], Gail V. W. Johnson [1] & Andrew P. Wojtovich [1] ✉

Hydrogen peroxide ($H_2O_2$) functions as a second messenger to signal metabolic distress through highly compartmentalized production in mitochondria. The dynamics of reactive oxygen species (ROS) generation and diffusion between mitochondrial compartments and into the cytosol govern oxidative stress responses and pathology, though these processes remain poorly understood. Here, we couple the $H_2O_2$ biosensor, HyPer7, with optogenetic stimulation of the ROS-generating protein KillerRed targeted into multiple mitochondrial microdomains. Single mitochondrial photogeneration of $H_2O_2$ demonstrates the spatiotemporal dynamics of ROS diffusion and transient hyperfusion of mitochondria due to ROS. This transient hyperfusion phenotype required mitochondrial fusion but not fission machinery. Measurement of microdomain-specific $H_2O_2$ diffusion kinetics reveals directionally selective diffusion through mitochondrial microdomains. All-optical generation and detection of physiologically-relevant concentrations of $H_2O_2$ between mitochondrial compartments provide a map of mitochondrial $H_2O_2$ diffusion dynamics in situ as a framework to understand the role of ROS in health and disease.

Mitochondrial reactive oxygen species (ROS) act as metabolic signals to coordinate bioenergetic state with cellular activity and adaptation. Under normal metabolic activity, the electron transport chain (ETC) produces low levels of ROS in the form of superoxide ($O_2^-$) which rapidly converts into hydrogen peroxide ($H_2O_2$) through dismutation[1]. Since $H_2O_2$ readily crosses membranes, it can act as a second messenger akin to $Ca^{2+}$. ROS accumulates at sub-toxic levels to regulate metabolic signaling. This basal redox tone is defined as oxidative eustress[2–4]. Pathological conditions increase ROS production in mitochondria beyond this basal state, termed oxidative distress. In disease, $H_2O_2$ accumulates and is released from mitochondria to aberrantly oxidize and impair cellular machinery, ultimately inducing apoptosis[5].

Years of investigating oxidative stress in disease have emphasized the role of ROS in driving cellular pathology[6–8], but surprisingly little is known about the "how" and "when" ROS generated in mitochondria drive global cellular changes. The exact concentration difference between healthy, metabolically signaling ROS and oxidative damage remains unknown. Once generated, ROS can diffuse between mitochondrial compartments and into the cytosol[9,10]. Moreover, the hypothesis that ROS releases from the mitochondrial matrix into the cytosol has growing evidence[11–14]. A deeper understanding of these processes would elucidate how ROS coordinates metabolic signaling and oxidative stress pathways.

[1]University of Rochester Medical Center, Department of Anesthesiology and Perioperative Medicine, 575 Elmwood Ave., Rochester, NY 14642Box 711/604USA. [2]University of Rochester Medical Center, Department of Pharmacology and Physiology, 575 Elmwood Ave., Rochester, NY 14642Box 711/604USA. [3]Research Group Epigenetics, Metabolism and Longevity, Research Institute for Farm Animal Biology (FBN), Dummerstorf 18196, Germany. ✉e-mail: andrew_wojtovich@urmc.rochester.edu

Mitochondrial $H_2O_2$ generation and diffusion are highly spatially organized and complex. Multiple sites along the ETC generate ROS at different rates, thereby forming distinct microdomains of $H_2O_2$ dynamics in mitochondria such as the matrix, intermembrane space (IMS), and along the outer mitochondrial membrane (OMM)[1,15–17]. Mitochondrial microdomains differ in the selectivity and rate of ROS scavenging[18], the propensity of $H_2O_2$ to diffuse due to distinct membrane porosity[19–22], and protein substrate crowding[23–25], which can alter downstream ROS signal propagation. Individual sites of ROS production in mitochondrial microdomains are linked to distinct diseases which further highlights the variation in local ROS production[6]. Beyond this biological complexity, methods to induce and measure ROS production have been historically restricted to drug treatments or genetic manipulation. These methods provide either spatial or temporal information, but not both. Recent advances have bypassed these constraints with targeted expression of $H_2O_2$-generating proteins such as the D-alanine oxidase (DAO) system[26]. These systems provide a high degree of spatial control of $H_2O_2$ production but at relatively slow timescales and with limited temporal regulation[9,11]. Furthermore, by producing $H_2O_2$ and not superoxide to mimic ROS generation at the ETC, these systems may not fully recapitulate the kinetic details of ROS diffusion dynamics.

Here, we employ an all-optical ROS generation and detection system to provide spatial and temporal control with endogenous dismutation in living cells. We use the superoxide ROS-generating protein tandem-KillerRed (KR)[27–29] and the $H_2O_2$ biosensor HyPer7[12] to measure ROS dynamics throughout mitochondrial microdomains. Spatially restricted ROS generation and monitoring in single mitochondria demonstrates the kinetics of ROS diffusion across subpopulations of mitochondria in a cell alongside the transient hyperfusion of mitochondria. Controlling for HyPer7 biosensor characteristics in each mitochondrial microdomain reveals that ROS in the IMS exhibits directionally selective diffusion into the matrix, where ROS favors the diffusion into the matrix over non-selectively diffusing through the OMM and into the matrix equally. Using this all-optical ROS generation and detection technique, we systematically generate a map of ROS diffusion kinetics into and out of each mitochondrial microdomain. We record the release of $H_2O_2$ from the matrix of single mitochondria into the cytosol and into surrounding mitochondria about one μm away in normal metabolic conditions. Together, these data suggest a model of how ROS generated in mitochondrial compartments can spread throughout a cell on a minute timescale.

## Results

### Optical control of ROS generation in single mitochondria in situ

ROS diffusion between mitochondria and the cytosol governs cellular responses to changes in metabolism, but the dynamics of ROS movement between these regions remain unknown. To monitor ROS dynamics in mitochondria, we first co-expressed the superoxide-generating protein tandem-KillerRed (KR) and $H_2O_2$ biosensor HyPer7, both targeted into the matrix of mitochondria in HEK293T cells and later in mouse embryonic fibroblasts (MEFs). The photosensitizer KR has a known superoxide quantum yield[30]. HyPer7 has been previously shown to be highly selective for $H_2O_2$ and sensitive enough to detect gradients of mitochondrial ROS in living organisms[11,12,31]. We reasoned that the coupling expression of these two proteins could record single mitochondrial ROS responses (Fig. 1a).

Single mitochondria in cells expressing matrix-targeted KR and HyPer7 were selectively photostimulated with 5 sec pulses of 561 nm light (88 μW). The HyPer7 and KR intensities of mitochondria in the cell were measured before and after photostimulation to track the diffusion of ROS between mitochondria over time. To account for mitochondrial movement across three-dimensional space, as well as fusion and fission dynamics, we developed a method to automatically segment and track single mitochondria for each frame of 20+ minute

recordings (5 sec between frames, >220 frames per experiment). The HyPer7 and KR intensity of each mitochondrion was normalized to its area in that frame to account for changes in the area over time. Additionally, the distance of each mitochondrion to the fixed spot stimulation point was calculated so that ROS diffusion could be measured in proximal and distal mitochondrial populations (approximately 5 μm and 15 μm away from spot stimulation, respectively). Together, this system allowed for the spatiotemporal monitoring of single mitochondrial ROS and morphological responses while limiting the effect of mitochondrial drift and size changes (Fig. 1b).

Spot photostimulation of KR led to distinct spatiotemporal ROS responses in single mitochondria as measured by HyPer7 (Fig. 1c, d). As expected, photostimulating KR led to rapid photobleaching as its chromophore oxidized as a product of photosensitization and ROS generation[27,32,33]. Mitochondria proximal to the spot stimulation had strongly photobleached KR and increased HyPer7 intensity, whereas mitochondria in the same cell which were distal to spot stimulation had a lower response of KR and HyPer7 (Supplementary Movie 1). Non-stimulated control mitochondria exhibited comparably similar responses to imaging as distal mitochondria (Supplementary Movie 2), indicating photostimulation of KR was constrained to mitochondria only proximal to the stimulation point. This suggested this technique could detect mitochondrial subpopulation responses to pulsed ROS generation.

Quantifying single mitochondrial subpopulation responses to ROS generation revealed an immediate, roughly 65% increase in HyPer7 intensity in mitochondria proximal to photostimulation (Fig. 1e, f). This indicated rapid dismutation of KR-generated superoxide into $H_2O_2$ and subsequent oxidation of the HyPer7 probe[11,12]. After the second photostimulation pulse on KR, proximal mitochondria were repeatedly oxidized to the same level compared to the first pulse despite HyPer7 not being fully reduced back toward baseline, indicating a lower photosensitization capacity for KR following initial photostimulation. This double photostimulation protocol began overwhelming matrix antioxidant machinery, as evinced by the decay kinetics of HyPer7 slowing after the second pulse ($K_{first}$ = 0.54 ± 0.07, $K_{second}$ = 0.085 ± 0.031 in $F/F_0$ per min) (Fig. 1g) without a change in final HyPer7 intensity (Fig. 1f). Together, these data demonstrated that this all-optical technique of pulse generating $H_2O_2$ reliably modeled oxidative distress with spatiotemporal precision.

### Tracking intracellular spread of spatially restricted mitochondrial ROS

To assess whether matrix-generated ROS could spread between mitochondria in a cell, we measured the oxidation of mitochondrial subpopulations proximal and distal to the photostimulation point. Surprisingly, we measured an increase in oxidation of distal mitochondria after a nearly five-minute delay following the second KR photostimulation pulse (roughly 15 min following the initial ROS pulse). This distal ROS spread correlated with rapid trafficking of oxidized mitochondria toward distal regions of the cell (Supplementary Movie 3), but mitochondrial motility across the cell was rarely visualized (~1–3% of total photostimulations) in our experimental timescale. This delayed increase in distal mitochondrial oxidation and trafficking was consistent with the idea that the second pulse of KR began overwhelming mitochondrial antioxidant machinery to subsequently impact the local redox environment outside mitochondria.

To probe this inter-mitochondrial diffusion of ROS further, we analyzed mitochondrial morphology before and after ROS photogeneration. Photostimulated mitochondria appeared to briefly elongate and increase contact with neighboring mitochondria (Fig. 1d, Supplementary Movie 4) rather than undergo direct trafficking throughout the cell. Frame-by-frame analysis of mitochondria revealed relatively stable number of mitochondria in proximal and distal

regions measured as a proportion to total (Fig. 1h). Contrastingly, mitochondria proximal to photostimulation exhibited transient waves of hyperfusion as measured by area and form factor (Fig. 1i, j), a metric which reports the complexity of mitochondrial shape. These waves of mitochondrial elongation continued throughout the duration of recording following an initial delay after the first ROS pulse, but slowed in frequency after the second pulse where distal mitochondria began similarly transiently increasing in area and form factor. Non-stimulated control mitochondria did not undergo measurable changes in morphology, further supporting that these changes were due to spatially restricted ROS generation through KR photostimulation.

These results were largely recapitulated in photostimulation experiments in MEF cells co-expressing matrix-targeted HyPer7 and KR (Supplementary Fig. 1A, B). Spot stimulation did minimally oxidize some distal mitochondria which fully decayed to baseline after both pulses of KR photostimulation. MEFs exhibited altered decay kinetics between ROS photogeneration pulses with a faster decay in the second pulse relative to the first ($K_{first} = 0.24 \pm 0.31$, $K_{second} = 1.73 \pm 0.34$ in $F/F_0$ per min) without a difference in final HyPer7 intensity (Supplementary Fig. 1C, D). This ROS photogeneration in MEFs also led to transient mitochondrial hyperfusion as measured by area and form factor (Supplementary Fig. 1E). Together, these results suggested ROS

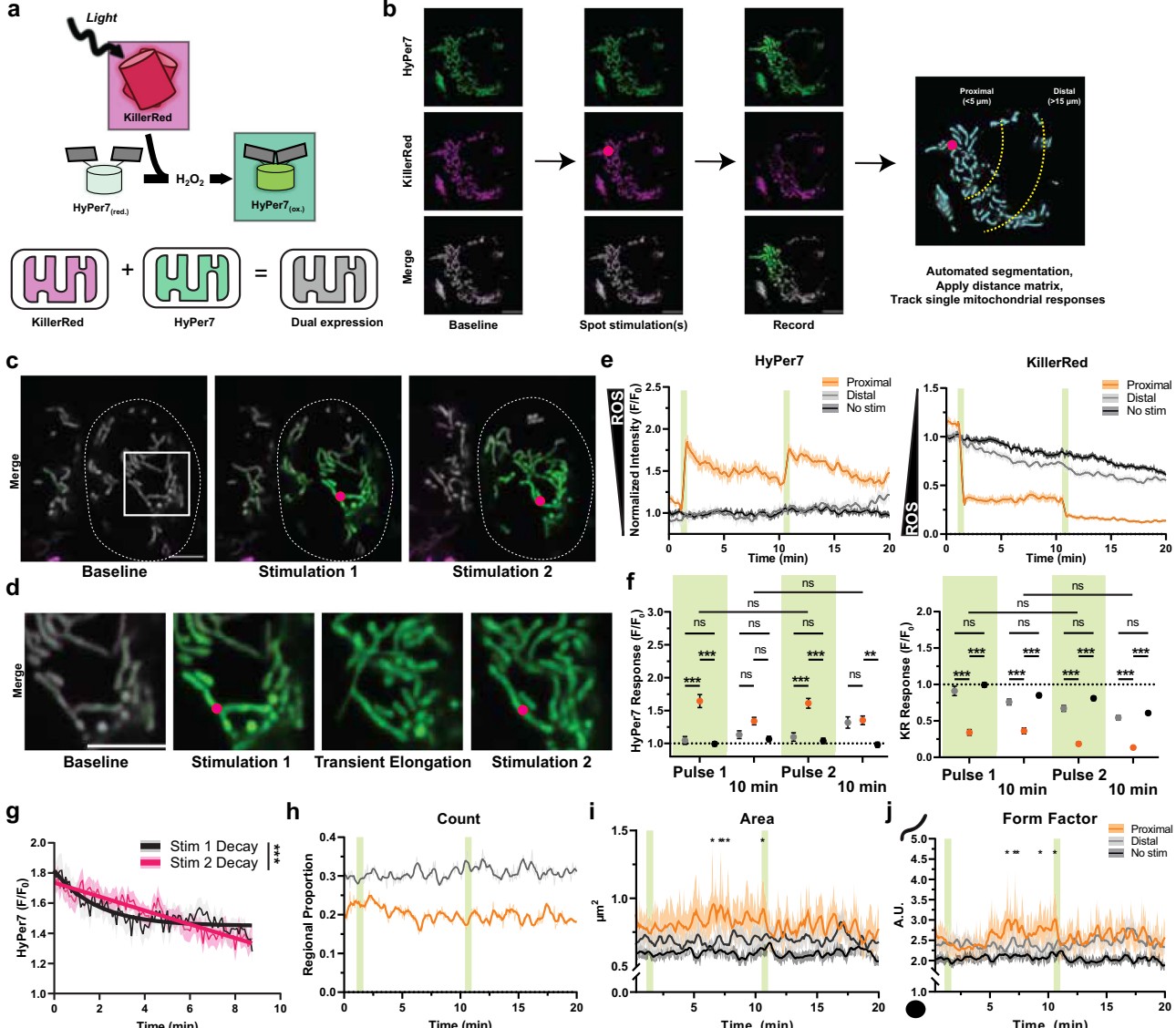

**Fig. 1 | Spatiotemporal manipulation and monitoring of ROS in single mitochondria. a** Schematic representation of all optical ROS generation and detection in mitochondria. **b** Imaging and analysis workflow to track spatiotemporal mitochondrial responses relative to single mitochondrial ROS photogeneration in HEK293T cells. **c** Representative images, where dotted line indicates a single cell expressing matrix-targeted HyPer7 and KR. Denoted circle indicates static spot stimulation used to pulse ROS photogeneration. **d** Inset from **c** highlighting transient elongation and contact of photostimulated mitochondria. **e** Single mitochondrial responses of HyPer7 and KR intensities normalized to area and baseline. **f** Comparison of HyPer7 and KR responses at first frame immediately following pulse 1, 10 min following first stimulation, immediately following pulse 2, and 10 min following second stimulation. Two-way ANOVA with Tukey post-hoc

multiple comparisons. **g** Decay of HyPer7 following stimulation 1 and 2 fitted with a nonlinear variable plateau followed by single phase decay, mean ± SEM. Decay compared with one-sided extra sum-of-squares F test. **h** Mitochondrial regional proportion of proximal and distal subpopulations relative to total in the cell over time. **i** Area of individual mitochondria in subpopulations over time through ROS photostimulation. Two-way ANOVA with Holm-Šídák's post-hoc multiple comparisons, mean ± SEM. **j** Form factor of individual mitochondria in subpopulations over time through ROS photostimulation. Two-way ANOVA with Tukey post-hoc multiple comparisons, mean ± SEM. Green bars indicate single frames of KR photostimulation. Mean ± SEM, $N = 39$ distal, 54 proximal, and 95 non-stimulated mitochondria per frame per group, on average. *$p < 0.05$, **$p < 0.01$, ***$p < 0.001$. Scalebars denote 5 μm.

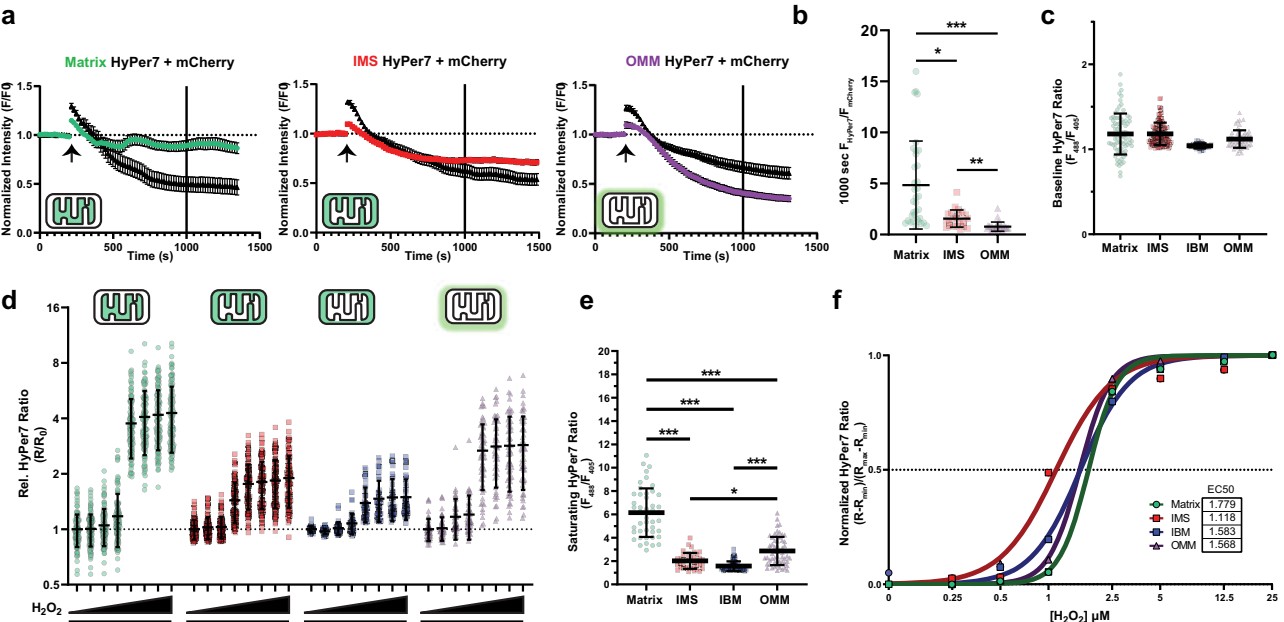

**Fig. 2 | Mitochondrial microdomain-specific responses to exogenous hydrogen peroxide. a** Normalized microdomain-targeted HyPer7 and mCherry signals per HEK293T cell relatively to baseline prior to treatment with digitonin and trypan blue. $N$ = 38 matrix-HyPer7; 32 IMS-HyPer7; and 34 OMM-HyPer7 cells per group, mean ± SEM. **b** Ratio of microdomain-targeted HyPer7 to mCherry per cell. $N$ = matrix-HyPer7: 35; IMS-HyPer7: 23; OMM-HyPer7: 28 cells, mean ± SD. One-way Kruskal-Wallis test with Dunn's post-hoc multiple comparisons correction. **c** Absolute HyPer7 ratio per microdomain at baseline. $N$ = matrix-HyPer7: 125; IMS-

HyPer7: 97; IBM-HyPer7: 43; OMM-HyPer7: 58 cells, mean ± SD. **d** Relative HyPer7 ratio response to increasing concentrations of hydrogen peroxide from 0.25 to 25 μM normalized to baseline per microdomain. N matches **c**, mean ± SD. **e** Absolute HyPer7 ratio per microdomain at saturating $H_2O_2$ exposure. One-way Kruskal-Wallis ANOVA with Dunn's post-hoc correction, N = matrix-HyPer7: 46; IMS-HyPer7: 38; IBM-HyPer7: 71; and OMM-HyPer7: 58 cells, mean ± SD. **f** Nonlinear sigmoidal fit of relative microdomain-specific HyPer7 responses to $H_2O_2$ after baseline and saturation normalization. *$p < 0.05$, **$p < 0.01$, ***$p < 0.001$.

transiently increases mitochondrial fusion, but the intra-cellular spread and decay of ROS may differ between cell types.

The photostimulation of KR can indirectly lead to mitochondrial elongation rather than through a ROS-mediated mechanism by depolarizing the inner-mitochondrial membrane. To test this, we combined the spectrally green mitochondrial membrane potential reporter Rhodamine123 (Rho123)[34] with the same photostimulation paradigms using 561 nm light (Supplementary Fig. 2A–C) to minimize the photobleaching that would occur with other red reporters. Photostimulation led to decreased Rho123 intensity which was not dependent on matrix KR. Since mitochondrial depolarization can lead to increased ROS formation[1], we further tested whether photostimulation of mitochondria without KR accumulate ROS and transiently elongates. Neither pulse of 561 nm light led to substantial increases in ROS in mitochondria as measured by matrix-HyPer7 (Supplementary Fig. 3A, B). Mitochondrial area and form factor were similarly unchanged (Supplementary Fig. 3C, D), consistent with the lack of detectable ROS formation due to changes in the mitochondrial membrane potential as a result of photostimulation.

### Mitochondrial fusion, and not fission, is required for ROS-mediated transient elongation

Mitochondrial elongation can occur through either diminished fission or heightened fusion activity. ROS has been previously shown to stimulate fission, leading to an accumulation of fragmented mitochondria which accumulate ROS, unable to restore redox homeostasis[35–37]. Altering the levels of mitochondrial morphology regulators such as the pro-fusion OPA1 and pro-fission DRP1 (encoded by the *DNML1* gene) proteins has been previously shown to modify ROS levels[35,38,39]. To test how lowering fusion or fission capacity regulates ROS dynamics, we generated HEK293T cells expressing shRNA for OPA1 or DNML1 to induce mitochondria with impaired fusion and fission, respectively,

and repeated single mitochondrion photostimulation experiments using matrix-HyPer7 and KR in these cells (Supplementary Fig. 4A–D). Puromycin selection induced a fragmented mitochondrial phenotype which was resistant to transient elongation after KR photostimulation. To control for this confound, we compared *OPA1* KD and *DNML1*-KD to shScrambled-expressing control cells. *OPA1*-KD and *DNML1*-KD mitochondria exhibited fragmented and hyperfused phenotypes, respectively, as expected (Supplementary Fig. 4E, F). Surprisingly, *OPA1*-KD mitochondria had diminished HyPer7 reduction capacity (Supplementary Fig. 4G) and a nearly complete lack of transient elongation (Supplementary Fig. 4E, F). Conversely, *DNML1*-KD cells had a heightened HyPer7 reduction capacity with rescued transient elongation. These results suggested that mitochondria dynamics can rapidly clear ROS by transient fusion, balanced in part by fission machinery maintaining mitochondrial morphology after hyperfusion.

### Mitochondrial microdomain-specific responses to exogenous $H_2O_2$

The ROS landscape varies greatly between mitochondrial compartments[1,15,16]. To understand ROS dynamics at each mitochondrial microdomain, we first confirmed proper microdomain-specific targeting of HyPer7. We targeted HyPer7 to three different microdomains: the matrix, intermembrane (IMS), and along the outer mitochondrial membrane (OMM). Cells co-expressing one of these three targeted constructs with cytosolic mCherry were imaged before and after addition with a mild treatment of digitonin to permeabilize membranes along with a fluorescent quencher, trypan blue, to confirm mitochondrial microdomain localization[40] (Fig. 2a). The fluorescence of microdomain-specific HyPer7 was compared to the reference fluorescent quenching of mCherry after over 15 min (1000 sec), when microdomain-specific differences in quenching rates would be apparent. As expected, quenching was dependent on targeting

location, whereby HyPer7 was quenched slowest in the matrix and fastest at the OMM, proportional to the number of membrane layers enclosing that compartment (Fig. 2b).

We validated this targeting approach by expressing microdomain-specific HyPer7 into a previously confirmed common background of *C. elegans* expressing mCherry targeted to the inter-boundary membrane (IBM) of mitochondria[31,41,42] (Supplementary Fig. 5A). The mitochondrial IBM represents the IMS, but without cristae[43,44]. Fluorescent proteins targeted to the IBM present as hollowed mitochondria when imaged at high resolution, thereby offering a visual reference standard to test microdomain-specific expression. Fluorescent line-scans of single mitochondria expressing microdomain-specific HyPer7 and IBM-mCherry further supported proper localization into the matrix, IMS, and OMM compartments based on the distance between compartment-specific HyPer7 and IBM-mCherry (Supplementary Fig. 5B, C).

With evidence of appropriate microdomain-specific targeting, we proceeded to identify how each compartment responded to exogenous treatments of $H_2O_2$. Whereas baseline steady-state levels of HyPer7 oxidation were largely unchanged between compartments (Fig. 2c), exogenous treatment of increasing $H_2O_2$ strongly affected HyPer7 responses (Fig. 2d). HyPer7 intensity at saturating levels of $H_2O_2$ significantly differed between microdomains (Fig. 2e). When normalized to microdomain-specific maxima, HyPer7 similarly responded to exogenous $H_2O_2$ treatment where the $EC_{50}$ in each compartment ranged between 1 to 2 $\mu M$ $H_2O_2$ with sharp saturation beyond 2.5 $\mu M$ (Fig. 2f). Given the similarity between the normalized $H_2O_2$-response curves between microdomains, interpolating the exogenous $H_2O_2$ concentration that induced a given HyPer7 intensity change per microdomain seemed reasonable.

HyPer7 is reportedly pH-resistant[12], but pH between mitochondrial microdomains can range up to a single unit[42]. To test whether the change in maximal HyPer7 response was dependent on microdomain-specific pH differences, we mutated the ROS-sensitive Cys121 residue to control for fluorescent changes independent of oxidation[12]. Baseline HyPer7(C121S) intensity further confirmed microdomain-specific targeting, as matrix-HyPer7(C121S) properly indicated a higher pH compared to IMS-HyPer7(C121S) (Supplementary Fig. 6A). Compared to HyPer7, the HyPer7(C121S) variant exhibited a dampened response in both the matrix and IMS to treatment with 100 $\mu M$ $H_2O_2$ and a heightened response to 40 mM $NH_4Cl$ to raise mitochondrial pH[45,46] (Supplementary Fig. 6B, C). Treatment with saturating levels of $H_2O_2$ after $NH_4Cl$ incubation still led to an increase in HyPer7 signal comparable to purely $H_2O_2$-treated cells (Supplementary Fig. 6D), indicating pH has a minimal effect on HyPer7 ROS responses. Overall, the different responses of HyPer7 biosensor activity between mitochondrial compartments were not explained by differences in pH.

We hypothesized HyPer7 could reveal distinct ROS clearance rates between microdomains because ROS scavenging and redox cycling proteins differ in the environments[47–52]. Surprisingly, HyPer7 decay rates did not differ between compartments and even between most $H_2O_2$ concentrations, decaying at ≈0.003 HyPer7 $R/R_0$ per sec (Supplementary Fig. 7A, B). Overall, these data show HyPer7 responds to low micromolar exogenously added $H_2O_2$, but a more endogenous system was needed to analyze ROS dynamics in a microdomain-specific context.

### Measuring endogenously produced ROS diffusion between microdomains

Respiratory complexes along the ETC are involved in signaling metabolic state and regulating behavior[31]. Complexes of the ETC form microdomains of local ROS environments whereby oxidation of key cysteine residues can impair organismal responses to nutrient and oxygen deprivation which can mimic early stages of neurodegeneration[31,53,54]. Studying these key oxidative events is difficult, in part due to the lack of available tools to selectively oxidize mitochondrial microdomains with spatiotemporal resolution. One option to generate ROS in specific microdomains is by using mitochondrial toxins to block electron transport through the ETC at defined sites. For example, complex I inhibition by rotenone produces matrix ROS and inhibits ROS from reverse electron transfer, while complex III inhibition with antimycin A generates ROS in the intermembrane space (IMS)[1,12,55–57] (Fig. 3a).

HyPer7 has been previously reported to be sensitive enough to detect microdomain-specific ROS generation by respiratory chain toxins[11]. We treated cells expressing microdomain-targeted HyPer7 with increasing concentrations of rotenone and antimycin A to monitor ROS diffusion from these defined generation sites throughout microdomains. Dual-color excitation HyPer7 intensity levels were compared to the red fluorescent dye TMRM, an indicator of mitochondrial membrane potential, $\Delta\Psi_m$. As rotenone and antimycin A block electron flow through the ETC, mitochondrial membrane potential drops as indicated by decreased TMRM signal which would limit NADPH regeneration and downstream antioxidant responses. Simultaneously, superoxide is generated at sites of ETC blockage, ultimately producing $H_2O_2$ and oxidizing HyPer7 (Fig. 3b).

Simultaneous measurement of $\Delta\Psi_m$ and ROS levels during acute ETC block revealed disparities in drug-mediated ROS diffusion through mitochondrial compartments. $\Delta\Psi_m$ was highly sensitive to rotenone and antimycin A, as expected, where 2 nM and 1 $\mu M$, respectively, were enough to significantly lower $\Delta\Psi_m$ (Fig. 3c and e). At higher concentrations, $\Delta\Psi_m$ was fully impaired and comparable to full $\Delta\Psi_m$ destabilization due to treatment with the protonophore FCCP. At our lowest concentration of 2 nM, rotenone significantly lowered matrix-HyPer7 oxidation compared to baseline (Fig. 3d), likely by inhibiting complex I and decreasing ETC-generated ROS, limiting ETC superoxide production[58,59]. Importantly, rotenone only oxidized matrix-HyPer7 and not IMS- or OMM-HyPer7 at any concentration (Fig. 3d). Under these experimental conditions, rotenone never induced ROS diffusion out of the matrix. Contrastingly, IMS-generated ROS through antimycin A diffused and oxidized both matrix- and OMM-HyPer7, but only once matrix-HyPer7 saturated at higher concentrations of antimycin A (Fig. 3g). This finding led us to estimate the amount of $H_2O_2$ equivalents generated in each compartment interpolated based on the microdomain-specific HyPer7 response to the $H_2O_2$ standard curve.

To obtain microdomain-specific response curves for interpolation, we nonlinearly fit the microdomain-specific HyPer7 $H_2O_2$ response curves from Fig. 2f. As described earlier, HyPer7 cannot be oxidized until more sensitive antioxidant machinery is sufficiently quenched in the compartment to which HyPer7 is localized. Since HyPer7 is tethered to specific compartments and unable to freely diffuse between them, we interpreted any change in HyPer7 fluorescence in a microdomain after a given stimulus as the consequence of reaching a steady-state level of ROS available for oxidation after diffusion and reaction with microdomain-specific antioxidant machinery. In other words, each microdomain-specific response curve of HyPer7 corresponds to how much ROS could oxidize HyPer7 at a given amount of exogenous $H_2O_2$ present in that microdomain after diffusion between compartments and antioxidant buffering. Therefore, microdomain-specific differences in HyPer7 fluorescence can be calibrated against the nonlinear fit to the HyPer7 response in the same microdomain to a given amount of exogenously added $H_2O_2$. In this way, the response of HyPer7 in the matrix following a stimulus can be compared to other stimuli by the amount of exogenous $H_2O_2$ equivalents needed to obtain the same change in matrix-HyPer7 fluorescence. Notably, this concentration does not denote the amount of $H_2O_2$ present, but instead, the equivalent concentration of $H_2O_2$ needed to be present in the cellular media (exogenously added) to reach the same level of change in HyPer7 fluorescence specific to that microdomain.

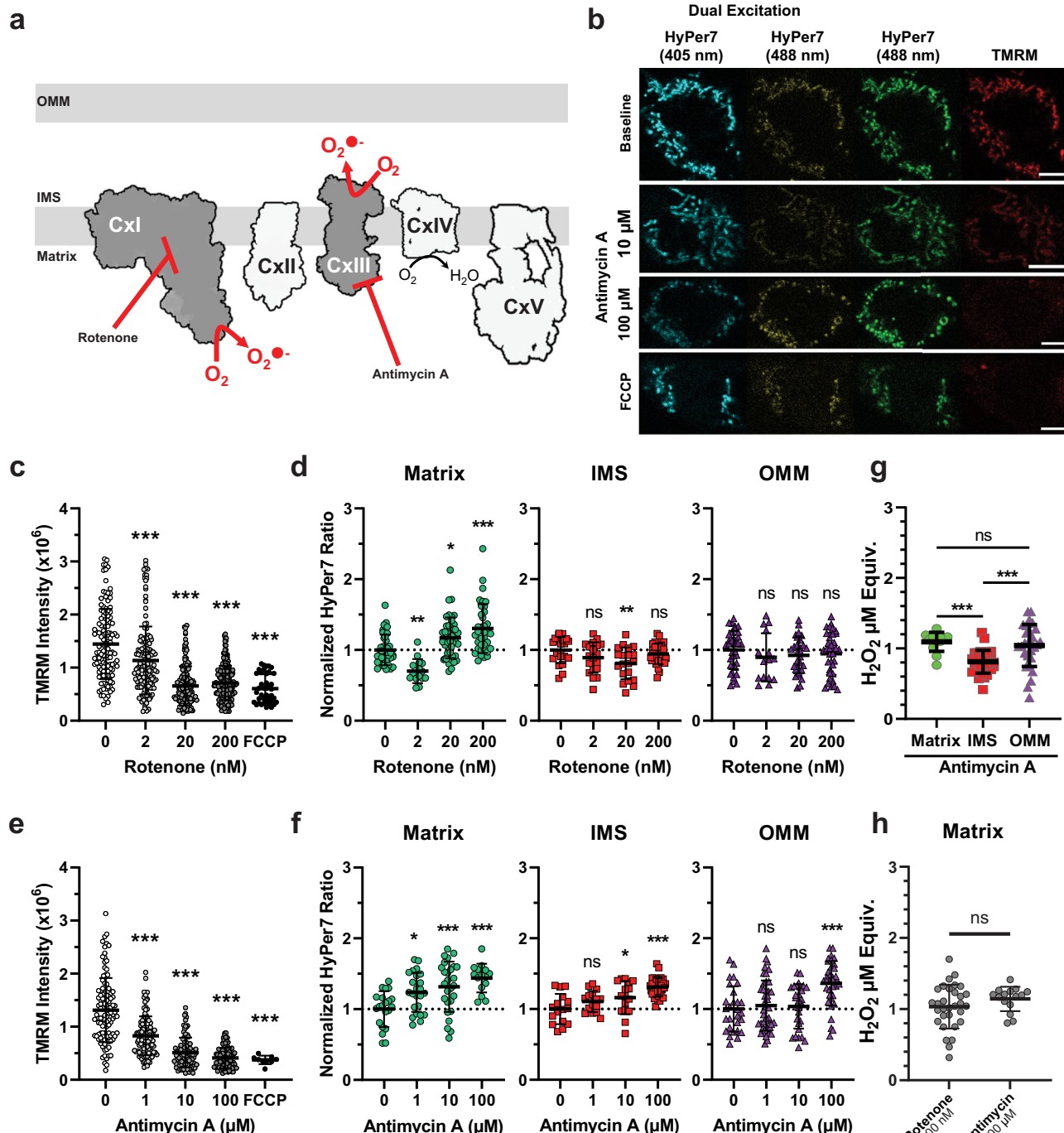

**Fig. 3 | Microdomain-specific ROS diffusion to canonical mitochondrial toxins.**
**a** Schematic of mitochondrial ETC toxins producing ROS at distinct sites.
**b** Representative images of dually excited HyPer7 and pseudo-simultaneous imaging of HyPer7 488 and TMRM in response to drug treatments in HEK293T cells. The scalebar denotes 5 μm. **c** TMRM responses of cells treated with increasing concentrations of rotenone and FCCP. One-way Kruskal-Wallis ANOVA with Dunn's post-hoc correction, $N = 40 – 193$ cells, mean ± SD. **d** Relative HyPer7 ratio responses to ROS produced at complex I through increasing rotenone concentrations. One-way Kruskal-Wallis ANOVA with Dunn's post-hoc correction, $N = 15 – 44$ cells, mean ± SD. **e** TMRM responses of cells treated with increasing concentrations of antimycin A and FCCP. One-way Kruskal–Wallis ANOVA with Dunn's post-hoc correction, $N = 9–114$ cells, mean ± SD. **f** Relative HyPer7 ratio responses to ROS produced at complex III through increasing antimycin A concentrations. Ordinary one-way ANOVA with Dunnett's post-hoc correction, $N = 15–43$ cells, mean ± SD. **g** Equivalents of exogenously added $H_2O_2$ needed to reach HyPer7 response to 100 μM antimycin A, normalized to microdomain-specific HyPer7 responses. One-way Kruskal-Wallis ANOVA with Dunn's post-hoc correction, $N =$ matrix-HyPer7: 13; IMS-HyPer7: 32; OMM-HyPer7: 31 cells, mean ± SD. **h** Equivalents of exogenously added $H_2O_2$ needed to reach HyPer7 responses due to maximal concentrations of rotenone and antimycin A in matrix-HyPer7 expressing cells. Two-tailed, Mann-Whitney test, mean ± SD, $N = 31$ rotenone, 15 antimycin A treated cells. $*p < 0.05$, $**p < 0.01$, $***p < 0.001$. Exact $N$ values are provided in the Source Data file.

Interpolating based on each microdomain-specific standard curve revealed that ETC toxins produced unequal amounts of $H_2O_2$ per microdomain. Antimycin A produced less $H_2O_2$ equivalents in the IMS ($0.81 \pm 0.16\,\mu M$) than that of the matrix ($1.09 \pm 0.13\,\mu M$) or OMM ($1.04 \pm 0.29\,\mu M$) (Fig. 3g). The maximal concentration used for rotenone generated $1.04 \pm 0.31\,\mu M$ in the matrix, roughly equal to antimycin A (Fig. 3h). Overall, these results demonstrated directionally selective diffusion between microdomains depending on the local antioxidant state.

### Mapping the diffusion of endogenously generated $H_2O_2$ between mitochondrial microdomains

Respiratory toxins induce prolonged generation of ROS, but physiological ROS can be transient. To induce transient ROS and probe whether ROS can differentially diffuse between mitochondrial microdomains, we co-expressed HyPer7 and KR tagged to either the matrix, IMS, or OMM. These nine experimental conditions encompass ROS generation and diffusion between every mitochondrial microdomain. Cells co-expressing microdomain-specific HyPer7 and KR were continually imaged for nearly 20 min through increasing durations of full-frame photostimulation of green light at a fixed irradiance ($0.42\,mW/mm^2$, Supplementary Fig. 8A, B). Each combination of HyPer7 and KR microdomain targeting revealed light dose-dependent effects of KR photostimulation which saturated at longer durations of 180 sec (Fig. 4a). Longer photostimulation times exhibited weaker HyPer7 responses, likely because of oxidative damage to the KR protein which prevented further photosensitization[27,32,33].

Interestingly, the rise times (time to half-maximal HyPer7 response, $T_{1/2}$), between compartments varied greatly (Fig. 4b). Matrix KR led to strong HyPer7 responses in all microdomains, but the matrix ($4.00 \pm 0.68$ sec) reached $T_{1/2}$ faster than the IMS and OMM ($7.38 \pm 1.02$ and $7.67 \pm 2.38$ sec, respectively). Consistent with our earlier results with antimycin A (Fig. 3f, g), ROS generated in the IMS with IMS-KR first led to a rapid increase in matrix-HyPer7 oxidation over IMS and OMM. The $T_{1/2}$ rise time from IMS-KR was equal in the matrix ($3.09 \pm 0.61$ sec) compared to the IMS ($3.05 \pm 0.45$ sec), with a significantly delayed increase at the OMM ($17.91 \pm 4.24$ sec). ROS generated at the OMM through photostimulation of OMM-KR more quickly diffused into the matrix ($T_{1/2}$ risetime $3.30 \pm 0.78$ sec) over the IMS ($7.46 \pm 2.14$ sec) and even OMM ($10.37 \pm 2.44$ sec). These data suggest ROS made in the matrix diffuses into the IMS, which rapidly releases into the OMM. ROS made in the IMS initially diffuses into the matrix before a delay, ultimately being released out of the mitochondria into the OMM. ROS made at the OMM, in contrast, diffuses into the IMS and matrix, detectably oxidizing matrix-HyPer7 first before a lag time where it eventually equilibrates and oxidizes IMS-HyPer7. Overall, these data support the directionally selective diffusion of ROS within mitochondria microdomains.

To account for microdomain-specific response curves of HyPer7 and to enable accurate cross-compartment kinetic comparisons, we interpolated the concentrations of exogenous $H_2O_2$ which would approximate the HyPer7 response to KR photostimulation per microdomain (Fig. 4c–e). Matrix KR photostimulation culminated in a greater maximal steady-state of $H_2O_2$ concentration of $1.03 \pm 0.02\,\mu M$ in the matrix versus $0.76 \pm 0.02$ and $0.81 \pm 0.04\,\mu M$ in the IMS and OMM, respectively. As photostimulation duration increased, the OMM-HyPer7 response increased to supersede the matrix response by 180 sec of photostimulation. At this maximal photostimulation duration, ROS generated in the IMS led to the highest $H_2O_2$ concentration at the OMM ($0.87 \pm 0.05\,\mu M$) compared to the IMS ($0.57 \pm 0.01\,\mu M$) and matrix ($0.78 \pm 0.02\,\mu M$), though the matrix remained significantly higher than the IMS. At maximal photostimulation duration of OMM-KR, this pattern was reversed whereby steady-state $H_2O_2$ was highest at the OMM ($1.12 \pm 0.05\,\mu M$) relative to the matrix ($0.82 \pm 0.02\,\mu M$) and IMS ($0.77 \pm 0.04\,\mu M$), as originally expected. While this data elucidated

initial directionality in diffusion between mitochondrial compartments, we further asked whether the decay kinetics of ROS in each compartment would reflect the ROS microenvironments present at each microdomain.

Superoxide dismutases (SODs) are endogenous enzymes that rapidly convert superoxide into $H_2O_2$. The levels of SODs vary between mitochondrial compartments, where the matrix contains SOD2 and the IMS and cytosol contain SOD1. To understand whether microdomain-specific SOD expression governs ROS dynamics and diffusion, we generated SOD1 and SOD2 knock-down cell lines (Supplementary Fig. 9A) and measured the rate of ROS diffusion between mitochondrial compartments. Interestingly, KD of either SOD increased basal ROS levels in some compartments as measured by baseline levels of HyPer7 oxidation without stimulation (Supplementary Fig. 9B, C). Consistent with this finding, $T_{1/2}$ risetimes of each microdomain between shScr-control expressing cells and SOD1/SOD2 KD cells changed considerably after SOD knockdown (Supplementary Fig. 9D, E). Knockdown of SOD1 led to mostly slower $T_{1/2}$ risetimes throughout microdomains regardless of ROS origin, but SOD2 KD led to hastened HyPer7 oxidation following ROS generation in the matrix and slowed HyPer7 oxidation following generation in the IMS. Overall, these findings supported the role of SODs in facilitating ROS dynamics and diffusion, but whether this is the direct result of altered superoxide dismutation into $H_2O_2$ or due to diminished cellular and mitochondrial health remains unknown[60].

### All-optical recording of ROS release kinetics from the matrix in single mitochondria

Redox scavenging and signaling machinery differ in expression among mitochondrial microdomains and the cytosol[61–64]. HyPer7 oxidation is partly dependent on the scavenging mechanisms competing for $H_2O_2$, whereas the decay rate of HyPer7 fluorescence following oxidation depends on the local redox machinery available to re-reduce HyPer7[11,12]. Having found differences in the approximate steady-state $H_2O_2$ levels produced by KR stimulation in each microdomain, we tested whether the HyPer7-KR system would be sensitive enough to detect differences in microdomain-specific rise and decay rates of $H_2O_2$ due to differing scavenging mechanisms. To accomplish this, we repeatedly induced ROS production in the matrix of single mitochondria through matrix-KR photostimulation and compared recordings of HyPer7 oxidation present in different mitochondrial microdomains. Given the technical limitations of slow light path switching between dual-color excitation of HyPer7 and single-color KR imaging, we resorted to simultaneous single-color excitation of HyPer7 (488 nm) and KR (561 nm) every 5 sec for over 20 min of recording followed by offline normalization of each individual mitochondrion's HyPer7 and KR intensity to its own area per frame to mitigate three-dimensional focal drift over time as in Fig. 1.

Broad details of ROS release from the matrix in mitochondria emerged when the oxidation of each microdomain-targeted HyPer7 was analyzed as fluorescent change over baseline. Spatially restricted ROS production through photostimulation of matrix-KR was confirmed by comparing the responses of mitochondrial populations expressing microdomain-targeted HyPer7 as a function of distance from a single pulse of 561 nm at a fixed point (Fig. 5a). As the distance between mitochondria and the fixed stimulation point increased, the resulting photobleaching of KR and subsequent oxidation of HyPer7 decreased. Matrix-HyPer7 failed to respond to matrix-KR photostimulation at approximately 12 $\mu m$, whereas IMS- and OMM-HyPer7 failed to respond at roughly 6 $\mu m$ and 1 $\mu m$, respectively. $H_2O_2$ generated by matrix-KR photostimulation produced decreasing HyPer7 responses as the targeted microdomains were further from the matrix, indicating the single pulse of ROS filtered through continuing antioxidant machinery per microdomain.

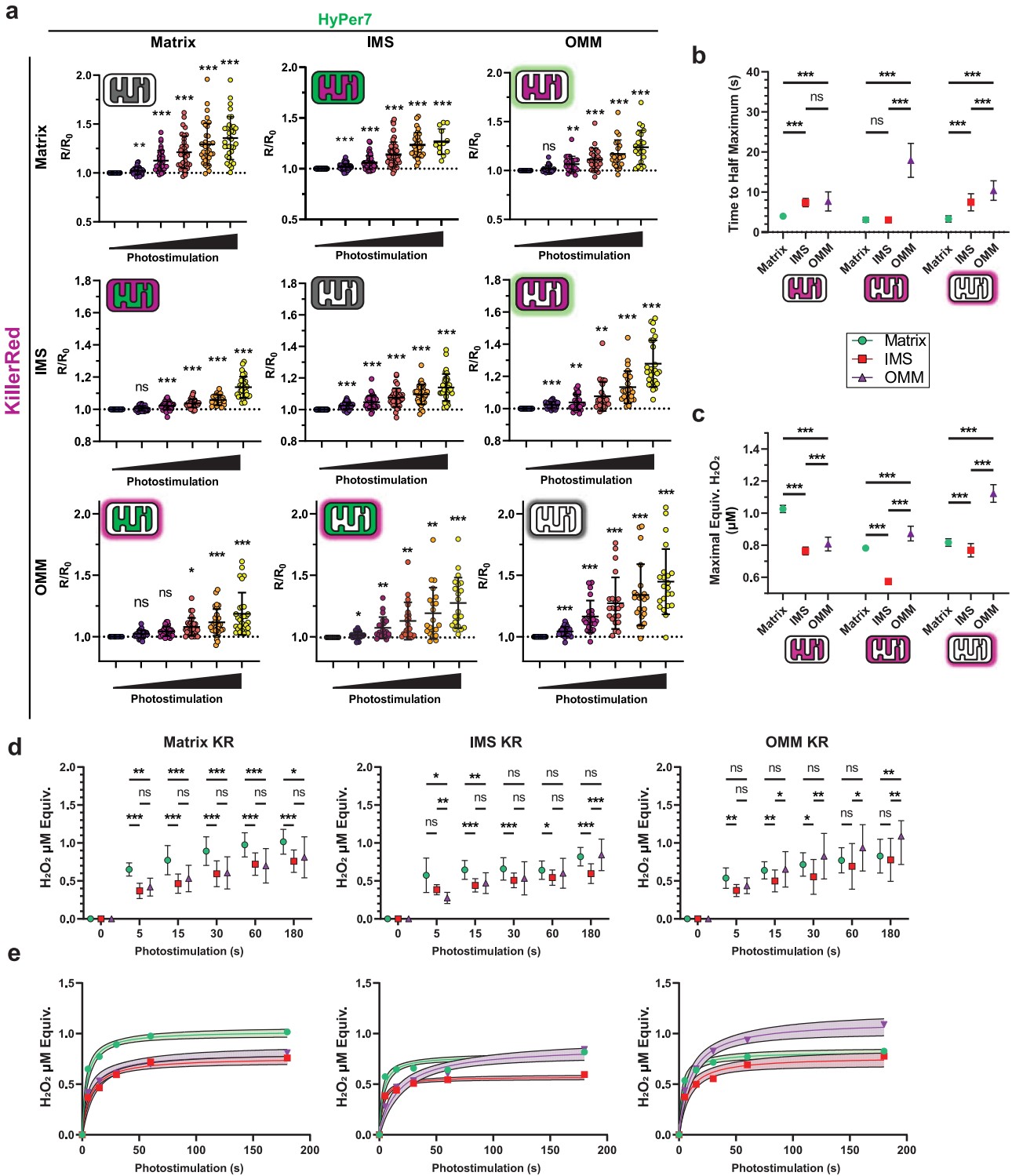

**Fig. 4 | Diffusion kinetics of mitochondrial microdomain-specific ROS. a** HyPer7 normalized ratio responses with KillerRed co-expressed in all mitochondrial microdomains. KillerRed photostimulation ranged from 0, 5, 15, 30, 60 and 180 sec. One-way, repeated measure, mixed-effects ANOVA with Geisser-Greenhouse correction and Dunnett post-hoc correction, $N = 21 – 55$ cells, mean ± SD. **b** Time to half maximum ($T_{1/2}$) for microdomain-targeted HyPer7 and KR experiments. Two-way ANOVA with Tukey post-hoc correction, $N = 21 – 55$ cells, mean ± SD. **c** Maximal interpolated steady-state $H_2O_2$ concentration of mitochondrial microdomains following microdomain-specific KR photogeneration of $H_2O_2$. Two-way ANOVA with Tukey post-hoc correction, $N = 21 – 55$ cells, mean ± SD. **d** Interpolated apparent $H_2O_2$ concentrations in each microdomain from A. Two-way, repeated measure, mixed-effects ANOVA with Geisser-Greenhouse correction and Tukey post-hoc correction, $N = 6 – 46$ cells, mean ± SD. **e** Interpolated apparent $H_2O_2$ concentrations as in B but depicted as nonlinear fit curves (all $R^2 > 0.6$) across all photostimulation timepoints. *$p < 0.05$, **$p < 0.01$, ***$p < 0.001$. Exact N values are provided in Source Data file.

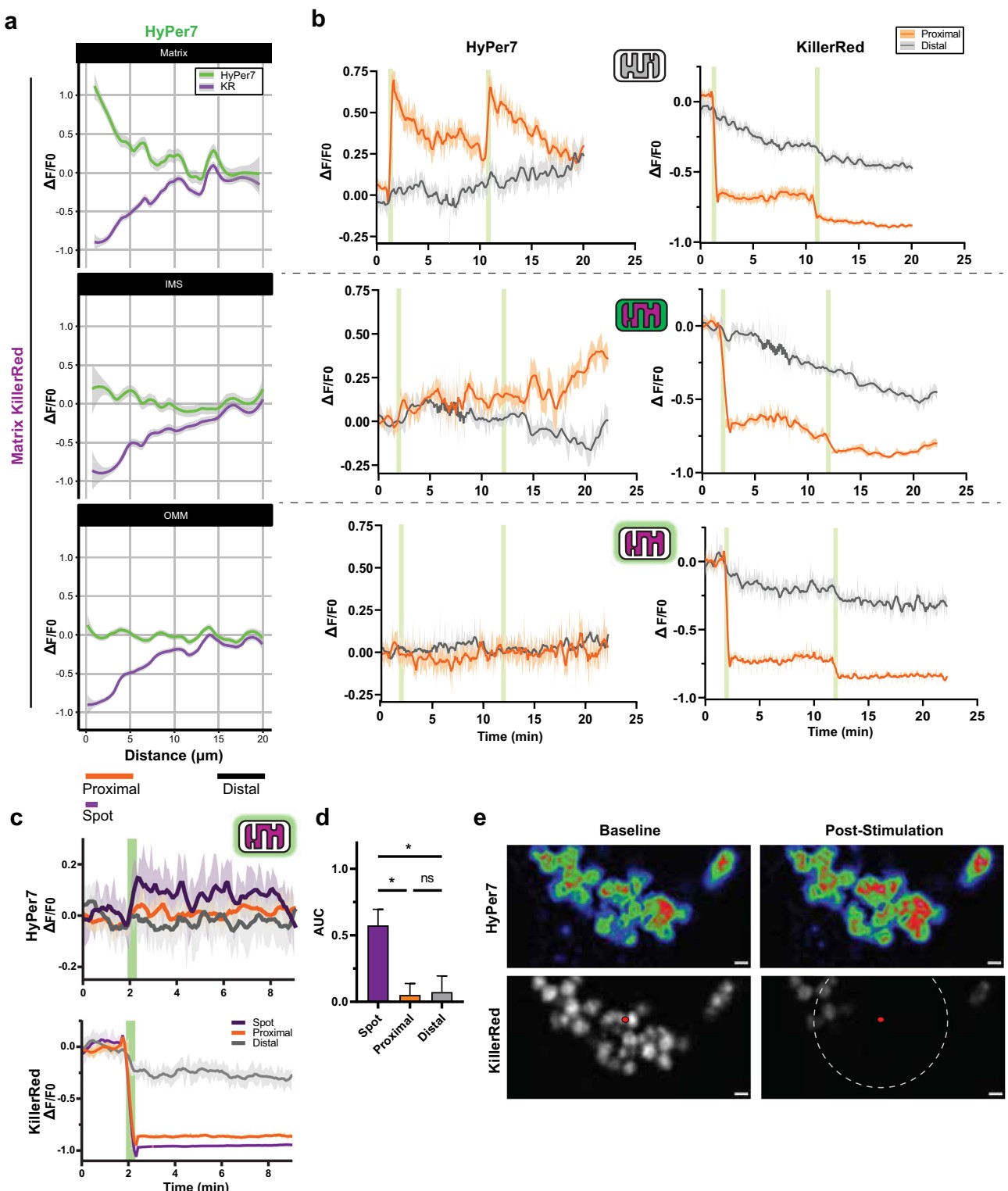

Contrary to whole-cell KR photostimulation (Fig. 4), spatially restricting acute photostimulation of KR revealed the variation in microdomain-specific responses over time. Matrix-HyPer7 exhibited repeatable stimulation, but the IMS-HyPer7 response was initially dampened until nearly five minutes after the second light pulse (Fig. 5b). We reasoned that this delayed increase in oxidation resulted from the saturation of IMS-specific antioxidants such as cytochrome c and the relatively lower amounts of available redox machinery at the IMS, resulting in ROS-induced ROS release (RIRR) in the IMS which was not apparent in the matrix[64–66]. Notably, this delayed increase in IMS

HyPer7 oxidation coincided with the second pulse decay of matrix HyPer7 and an increase in mitochondrial motility. Interestingly, neither pulse of matrix KR photostimulation led to increased OMM-HyPer7 responses, indicating this acute, spatially restricted ROS generation paradigm did not overwhelm antioxidant machinery enough to oxidize OMM-HyPer7 and measurably alter its fluorescence.

To test whether we could detect release of matrix ROS into the OMM in single mitochondria, we increased the duration of photostimulation pulse from 5 sec to 30 sec. Increasing the photostimulation time and further subdividing mitochondria proximal to the spot

**Fig. 5 | Dynamics of single mitochondrial matrix ROS release. a** Fluorescent change over baseline of single mitochondria in HEK293T cells co-expressing various microdomain-targeted HyPer7 with matrix-KillerRed. Data taken from 12 frames (one minute) following the first 5 sec photostimulation pulse and displayed as a function of distance from the spot stimulation point. Data expressed as mean ± 95% CI, $N$ = 192 matrix-HyPer7, 357 IMS-HyPer7, and 120 OMM-HyPer7 mitochondria, on average. **b** Average traces of fluorescent change over baseline of single mitochondria co-expressing microdomain-targeted HyPer7 and matrix-KillerRed following repeated photostimulation. Mitochondrial populations were separated based on distance to spot stimulation, where proximal and distal roughly approximates under 5 μm and over 15 μm, respectively. Data expressed as smoothed mean ± SEM. $N$ = matrix-HyPer7: 39 proximal and 49 distal; IMS-HyPer7: 32 proximal and 34 distal; OMM-HyPer7: 53 proximal and 31 distal mitochondria, on average. **c** Fluorescent change over baseline of single mitochondria expressing OMM-HyPer7 and matrix-KillerRed following a single, longer duration (30 sec) photostimulation pulse of KillerRed. Mitochondrial populations were separated as proximal and distal, but also included a "spot" subpopulation of mitochondria within 1.5 μm of the stimulation point. Data expressed as smoothed mean ± SEM, $N$ = 7, 27, and 52 individual mitochondria for spot, distal, and proximal mitochondrial populations per frame, on average. **d** Area under the curve analysis of C. Data expressed as mean ± SEM, one-way ANOVA with Tukey post-hoc correction, $N$ = 4 independent experiments. **e** Representative images of mitochondria in a cell expressing OMM-HyPer7 (rainbow LUT applied) and matrix-KillerRed (grayscale) before and after photostimulation. Spot stimulation point depicted as a red circle, 5 μm surrounding area as a dashed white circle. Scalebar denotes 1 μm. *$p < 0.05$, **$p < 0.01$, ***$p < 0.001$. Note: Matrix-HyPer7 data in **b**) taken from Fig. 1 and re-expressed as dF/F.

stimulation point to within 1.5 μm of the spot stimulation point (denoted as "spot," compared to proximal of less than 5 μm and distal as over 15 μm away) revealed a marked increase in OMM-HyPer7 oxidation that remained for nearly 10 min (Fig. 5c). Area under the curve (AUC) analysis revealed this prolonged stimulus oxidized OMM-HyPer7 in a highly spatially restricted manner (Fig. 5d, e). These data suggest that coupling expression of microdomain-targeted HyPer7 and KR with single mitochondrial photostimulation can reveal the spatiotemporal diffusion kinetics of endogenously produced $H_2O_2$ between compartments and measure the release of matrix ROS into the cytosol without other alterations.

## Discussion

Here, we use microdomain-targeted expression of the $H_2O_2$ biosensor HyPer7 and ROS photogenerator KillerRed to measure the kinetics of ROS generation and diffusion throughout mitochondrial microdomains. One limitation of this study is the inability to directly control for microdomain-specific differences in volume or microdomain insertion of targeted proteins. Given the apparent differences in microdomain-specific HyPer7 responses, controlling for these differences was crucial for inter-microdomain comparison. To account for this, all intensity measurements described here involve multiple measures of intra-experimental normalization. For example, HyPer7 ratiometric intensities were used in most comparisons which mitigated differences in biosensor expression and insertion. During single mitochondrial imaging experiments, single excitation HyPer7 intensities per mitochondrion were normalized to that same mitochondrion's area in each frame, which similarly limited expression-based confounds. Cross-compartment, all-optical perturbation of ROS using HyPer7 and KR were normalized first to each individual cell's KR expression, and then by measuring ratiometric HyPer7 intensities relative to the baseline of the same cell following photostimulation. Moreover, interpolated exogenous equivalents of $H_2O_2$ concentrations were based on microdomain-specific responses, further controlling for any apparent differences. Together with multiple factor assessment of microdomain-specific targeting and HyPer7 pH dependence, these methods attempted to account for inherent bias in technical approaches. It remains possible, though unlikely, that fundamental microdomain-specific changes in biosensor responses bias our results despite these normalization approaches. Thus, the interpretation of the resulting exogenous equivalents of $H_2O_2$ concentration we report herein should be carefully considered given the technical and biological complexities of measuring $H_2O_2$ dynamics in situ[1,67]. Lastly, it remains possible that part of the HyPer7 responses detected here are due to direct interaction with superoxide produced by KR. The likelihood of this is low, however, since HyPer7 has a low sensitivity to superoxide compared to $H_2O_2$[12] and considering superoxide would rapidly undergo dismutation into $H_2O_2$ in living cells likely before oxidizing HyPer7.

We use microdomain-specific standard curves of exogenously added $H_2O_2$ to compare HyPer7 responses due to toxins or KR photogeneration of ROS between microdomains. Since $H_2O_2$ diffuses through membranes[19,22,68,69] and local redox scavengers such as the peroxiredoxins shift HyPer7 responses by competing for $H_2O_2$[11,12], we consider these microdomain-specific responses a reflection of apparent redox-state in that compartment. In other words, an increase in HyPer7 signal means the antioxidative machinery in that compartment could no longer prevent ROS from oxidizing proteins such as HyPer7, reducing any oxidized thiols, and/or dismutase activity in that compartment facilitated $H_2O_2$ production. Moreover, antioxidant buffering systems such as the peroxiredoxin and thioredoxin systems responding to $H_2O_2$ would continually diffuse and equilibrate between cellular compartments during ROS clearance[11]. Therefore, a lack of HyPer7 oxidation does not necessarily indicate the lack of ROS diffusion[70,71]. This is best exemplified in the IMS, where the relatively lower concentration of SOD compared to the matrix and cytosol, but with an increase in cytochrome c antioxidant levels, is predicted to sequester superoxide prior to dismutation into $H_2O_2$[47,60,64,72]. These differences could ultimately limit $H_2O_2$ oxidation of HyPer7 and explain the consistently lower apparent $H_2O_2$ levels detected by HyPer7 in the IMS in our experiments.

Moreover, since the concentration of exogenously added $H_2O_2$ is likely an over-estimate of intracellular and subcellular accumulation, interpolating local concentrations of $H_2O_2$ based on HyPer7 measurements can only provide an upper-bound estimate. Other studies have modeled a 390 to 650-fold difference between extracellular and intracellular $H_2O_2$[73,74] and calculated a basal steady-state of 2 to 4 nM in mitochondria[9]. Our experiments never produce a steady-state change approximating greater than the equivalent of treatment with 2 μM $H_2O_2$. This suggests the apparent mitochondrial accumulation of $H_2O_2$ corresponded to between a 3 to 5 nM increase above this baseline to 5 to 9 nM, though this number is likely specific to both cell type and metabolic conditions. Acute ROS generation experiments using toxins and KR reported here never caused overt cell death during imaging and had only begun to overload redox machinery between compartments. Therefore, we predict our estimated 3 to 5 nM increase in mitochondrial $H_2O_2$ delineates the transition from oxidative eustress toward distress. Higher levels of ROS production induce mitochondrial aggregation, dysfunction, mitophagy, and ultimately cell death[27,32,41,75–78]. Future studies in alternative models, scavenging, and metabolic or disease conditions may further investigate the responses to changes in ROS.

Previous reports have indicated ROS-mediated hyperfusion acts to prevent pathological accumulations of ROS in the mitochondrial network[79–82]. Here, we show that spatiotemporally limited pulsing of matrix ROS generation induced transient mitochondrial hyperfusion and intracellular mixing of ROS through the active movement of mitochondria. The first pulse of matrix ROS induced local hyperfusion. The second pulse induced less transient hyperfusion, but also

increased ROS levels found in distal mitochondria relatively far from the site of ROS photogeneration. This network ROS spread was delayed by several minutes following the second pulse of ROS itself and was limited by cell-type. However, in human cells, this inter-network spread occurred during the decay of matrix HyPer7 and during the increase in IMS HyPer7 oxidation, indicating ROS was being released from the matrix. We speculate this inter-mitochondrial spread of ROS acts to dilute ROS contents over a greater population of mitochondria, both decreasing the damage of any singular mito-chondrion but also hastening the global cell response to oxidative distress. In support of this theory, cells with impaired fusion (*OPA1* KD) had a slower decay speed of oxidized HyPer7, suggesting impaired antioxidant capacity, along with nearly complete inhibition of tran-sient elongation. In cells with impaired fission (*DNML1* KD encoding DRP1), oxidized HyPer7 had more rapid decay along with greater oxi-dation of distal mitochondria, suggesting heightened antioxidant defense and greater diffusion. Fission-deficient cells also had more transient elongation events as measured by both area and form factor. Overall, this work impairing mitochondrial fusion/fission machinery suggest that ROS-mediated transient elongation events require fusion and at the very least correlate with hastened ROS clearance in mito-chondria. An important extension of this work would be investigating mitochondrial ROS spread in cells with unique morphology, such as neurons, where mitochondria can be trafficked over relatively large distances.

Whether this increase in HyPer7 oxidation in distal mitochondria is produced by the inter-mitochondrial exchange of $H_2O_2$ or oxidized HyPer7 is unknown, but either possibility supports inter-mitochondrial content exchange to ostensibly lower any single mitochondrion's damage due to ROS. Given that the spatial positioning of mitochondria influences subcellular redox status[83], this transient increase in fusion followed by mitochondrial motility could work to prevent aberrant mtDNA and protein oxidation in response to mild ROS stimulus[80,84]. We note that the trafficking of ROS-stimulated mitochondria from the periphery to opposite sides of the cell did occasionally occur during our experimental timescales, suggesting that rapid trafficking of ROS-damaged mitochondria across the cell is a potential avenue for mito-chondrial ROS signaling open for further study.

Many studies have used rotenone and antimycin A to generate and accumulate ROS to study downstream physiological and patho-logical effects. Our acute treatments of rotenone and antimycin A across low to high doses generated steady-state HyPer7 oxidation levels which approximated treatments of under $2\,\mu M$ $H_2O_2$. Notably, care should be taken not to equate the treatment of 10 or more $\mu M$ $H_2O_2$ with the short-term treatment of rotenone or antimycin A given these differences. Our lowest dosage of rotenone (2 nM) lowered matrix-HyPer7 oxidation, consistent with the theory that low dose complex I inhibition decreases ROS and provides physiological bene-fits that extend lifespan[85]. Across all dosages, however, rotenone did not lead to oxidized HyPer7 in the IMS and OMM. As said previously, this lack of HyPer7 oxidation does not necessarily indicate the lack of ROS spread, but instead, the antioxidant systems in these compart-ments adequately cleared any diffusing ROS and prevented the oxi-dation of HyPer7. In contrast, the generation of ROS in the matrix through matrix-KR photostimulation led to equal increase in HyPer7 oxidation in the IMS and OMM. This generally agrees with the "flood-gate" model of ROS dynamics[70,71], where a prolonged ROS stimulus (e.g., rotenone) can be cleared over time, but a sudden larger insult of ROS (e.g., pulsed KR stimulation) overwhelms redox machinery and leads to diffusion between compartments and ROS release into the cytosol.

Unlike matrix ROS, IMS ROS diffuses in a directionally selective manner depending on concentration. Increasing dosages of antimycin A induced increasing HyPer7 oxidation in the matrix before being released from mitochondria and detectably diffusing into the OMM.

Using KR as an optically-controlled photosensitizer allowed finer temporal control of this directional selectivity, where the lowest photostimulation pulse induced mostly inward ROS diffusion to the matrix. Lengthening photostimulation duration led to saturation of matrix HyPer7 and subsequent OMM-HyPer7 oxidation. It remains to be seen if a biosensor can detect basal differences in ROS between mitochondrial compartments as predicted based on physiological site-specific production of ROS along the ETC.

Considerable debate has focused on whether $H_2O_2$ generated within the mitochondrial matrix can be released into and impact other cellular compartments[12]. Here, we maintained constant meta-bolic conditions of relatively low, near-physiological 3–5 mM glucose which protected mitochondrial morphology and limited down-stream effects of hyperglycemia such as ROS accumulation during imaging experiments[86–89] compared to the more common 10-25 mM concentrations seen in culture media in addition to other fuel sour-ces (as in[11]). Using systematic, all-optical cross-compartment detec-tion and generation of $H_2O_2$, we uncover the spatiotemporal release dynamics of matrix-generated ROS into the cytosol without meta-bolic or redox machinery perturbation. As our experiments with OMM-KR indicate, ROS generated on the outer membrane of mito-chondria is rapidly internalized in the matrix and can oxidize HyPer7, even during the shortest photostimulation pulse of KR. We speculate that the ROS generated on the OMM is detected earliest in the matrix before the IMS due to the nature of our IMS localization tag being present in the cristae and IBM. ROS generated on the OMM may diffuse through the OMM into the IBM and directly into the matrix or into the IBM and then into the cristae, which is considerably thicker and filled with more HyPer7 molecules. Thus, it is not that the ROS bypasses the IMS altogether, but rather that the ROS takes longer to oxidize HyPer7 to detectably change fluorescence. Our single mito-chondrial photostimulation experiments using matrix-KR are con-sistent with this idea, where strong pulses of matrix ROS can be released and diffused between mitochondria across roughly 1-2 µm to oxidize matrix-HyPer7. This is consistent with the estimated life-time diffusion of $H_2O_2$ in the cytosol of roughly 1 ms spreading over about 1 µm in the cytosol[90,91].

Together, these data (summarized in Supplementary Fig. 10) reveal the diffusion kinetic and directionality details of endogenously produced ROS between mitochondrial compartments in living mammalian cells. Beyond these parameters, we suggest a model whereby ROS generation causes prolonged effects on mitochondrial morphology and ROS steady-state levels due to local ROS release from mitochondria into the cytosol, where mitochondrial hyperfu-sion and motility can lead to cell-wide changes in redox state. These findings further support how understanding ROS dynamics between mitochondrial microdomains can reveal how ROS can act as a metabolic signal which propagates within the mitochondrial net-work, influencing cell-wide physiological responses over short timescales.

## Methods
### Plasmid construction
Matrix- and IMS-HyPer7 were originally received as a gift from Vsevo-lod V. Belousov (Addgene #136470 and 136469). KR from Evrogen (#FP963, pArrestRed) was subcloned in another study[41]. To alter tags and vectors for use in cells and worms in this study, constructed were subcloned using in vivo assembly (IVA)[92]. HyPer7(C121S) was mutated using point mutagenesis. All primers used are supplied in Supple-mentary Data 1. Microdomain targeting was accomplished using N-terminal fusions of tags to the matrix using 2xCOX8A (tag corre-sponds to Cox8A N-terminal residues 1-25, from Addgene 136470), the IMS using SMAC (residues 1-59, from Addgene 136469), the IBM using IMMT (residues 1–187[41]), and the OMM using TOMM20 (residues 1-55, from Addgene 66753).

## Cell culture and transfection

HEK293T cells (System Biosciences) and mouse embryonic fibroblasts (MEFs, ATCC) and were grown at 37 °C with 5% $CO_2$ in DMEM medium (glucose 4.5 g/L) supplemented with 10% fetal bovine serum (FBS), 2 mM GlutaMax (ThermoFisher), and penicillin/streptomycin and plated onto MatTek 35 mm glass-bottomed dishes (20 mm, No. 1.5) at approximately 200,000 cells ml$^{-1}$ two days before imaging and one day before transfection. For TMRM and drug experiments, 50,000 cells ml$^{-1}$ were plated into 8-well chambered No. 1.5 coverslips (Ibidi). DNA transfections for HEK cells used PolyJet (SignaGen Laboratories), whereas MEF cells used TransIT-X (Mirus) following manufacturer protocol with varied DNA concentrations. For both cell types, each plate was transfected using 0.15 μg of each microdomain-targeted construct and brought to 1 μg with empty pcDNA3.1+ vector.

## Stable cell line generation and transient transfection selection

Lentiviruses were packaged in HEK293T cells. In brief, shRNA constructs (Addgene #102975, 102976, 8453, 99385, and 191949) were co-transfected into HEK293T cells with PSP and VSVG viral coat and packaging proteins. HEK293T cells were kept at 33 °C with 5% $CO_2$ for 36 h. After 36 h, the virus-containing media was collected and filtered with a 0.45 μm filter. Viral pellets were collected in 1% FBS media and kept at −80 °C for later use. For viral transduction, the thawed virus was added to HEK293T plated the day prior. Media was replaced on the cells with 10 ml of DMEM media supplemented with 10% FBS, 1% Pen-Strep, and 0.6 μg/ml Puromycin. Cells were carefully passaged while changing the media as necessary over 2 weeks. Puromycin selection was stopped two days prior to imaging. For mitochondrial morphology experiments using shScr, shOPA1, and shDNML1/DRP1 cells were selected for 3 days after transient transfection using Puromycin prior to imaging followed by RT-qPCR to confirm knock-down.

## qRT-PCR

RNA was extracted from transduced HEK293T cells using the E.Z.N.A Total RNA Kit I (Omega Bio-tek R6834-02) per the manufacturer's protocol. To create a cDNA library, 300 ng of RNA was used with the Verso cDNA Synthesis Kit (Thermo Scientific AB-1453/B) per the manufacturer's protocol. Real-time PCR was performed using 2x Universal SYBR Green Fast qPCR Mix (Abclonal RK21203) on a CFX Duet Real-Time PCR System (Biorad). Primers are included in Supplementary Table 1. Relative expression of target transcripts was normalized to HPRT as a housekeeping control using the $2^{-\Delta\Delta Ct}$ method.

## Live cell imaging

Transfected cells had media replaced with prewarmed HBSS imaging buffer composed of (in mM): 5.33 KCl, 0.44 $KH_2PO_4$, 4.16 $NaHCO_3$, 137.9 NaCl, 0.33 $Na_2HPO_4$ supplemented with 20 HEPES and 3-5 glucose for at least 20 min prior to imaging. For some experiments, TMRM (25 nM) or Rhodamine123 (10 nM) was added to the imaging media. For respiratory toxin experiments, varying concentrations of rotenone (Sigma) and antimycin A (Sigma) were included in the imaging buffer. Cells treated with rotenone and antimycin A were incubated at 37 °C with 5% $CO_2$ for 30 and 60 min, respectively, prior to imaging.

Cells were imaged using an inverted Nikon A1R HD microscope with 405, 488, and 561 nm laser lines with a CFI Apochromat TIRF 60x oil objective using Nikon NIS-Elements AR version 5.42.01 build 1793. HyPer7 was imaged in dual excitation, single emission setup with 405 and 488 nm excitation unless otherwise stated. KillerRed was imaged using the lowest possible 561 nm excitation. Images taken using resonant scanning used at least 2x line averaging which was kept constant for each experiment. Galvano scanning was used for single mitochondrial imaging with 11.67x optical zoom (19.3 px/μm). To ensure minimal drift, samples were allowed to equilibrate on the microscope for 5–10 min prior to imaging onset.

## KillerRed activation

Activation of KillerRed in the entire field of view used a white light Sola light engine filtered through a TexasRed filter (30.5%, 0.42 mW/mm$^2$). Spot stimulation of KillerRed was targeted with a 561 nm laser (30% power, 88 μW, 30 Hz, 100% duty cycle for 5 sec stimulation).

## Mitochondrial topology

Cells expressing various microdomain-targeted HyPer7 constructs were co-transfected with cytosolic mCherry for use as a comparison standard between quenching experiments. HyPer7 and mCherry were imaged for 3 min baseline in imaging buffer, followed by two washes and replacement of media with prewarmed intracellular buffer (in mM): 10 NaCl, 130 KCl, 1 $MgCl_2$, 1 $KH_2PO_4$, 2 succinic acid, and 20 HEPES (pH 7.4) supplemented with 0.5 mg/ml trypan blue and 20 μM digitonin. Experiments with movement artifacts due to media replacement were discarded so single-cell traces of HyPer7 and mCherry signal could be obtained and compared to the same-cell baseline throughout the entire recording. Cells that did not respond to digitonin were not analyzed.

## HyPer7 responses

Hydrogen peroxide was diluted with imaging buffer and kept on ice. Baseline measurements of HyPer7 steady-state in each microdomain were measured following baseline recording where cold imaging buffer was added to mock $H_2O_2$ treatment. Increasing concentrations of $H_2O_2$ were applied to cells and imaged for at least 3 min before treatment with the next concentration. For decay kinetic experiments, HyPer7 responses to some concentrations were imaged for 10-20 min prior to the next addition of $H_2O_2$. Saturated responses of microdomain-targeted HyPer7 were taken from 100 μM additions of $H_2O_2$ either independently or at the end of the increasing $H_2O_2$ concentration curves. $H_2O_2$ concentrations were interpolated from averages of microdomain-specific responses of HyPer7 based on a sigmoidal curve where $R^2 > 0.99$ for all curves. Interpolation was limited to cells that exhibited a response to either toxin- or photostimulation-induced HyPer7 oxidation and did not include cells that decreased below baseline.

## Single mitochondrial photostimulation

Single cells brightly expressing microdomain-targeted HyPer7 and KillerRed were chosen for spot stimulation and single mitochondrial analysis. A pre-determined spot in the cell was targeted for repeated stimulation and remained unmoved during imaging. The XY coordinate of the spot was marked for later analysis. Following 1.5–3 min baseline recording, two iterations of a 5 sec spot stimulation followed by an 8–10 min recording were imaged. Since imaging KillerRed also leads to slight ROS photogeneration, the lowest possible 561 nm laser power (0.27% power, 2.2 μW) was used which still enabled single-mitochondrial resolution. Some videos had the first 3–4 frames (15–20 sec) removed due to substantial mitochondrial drift immediately after imaging onset. No images underwent photobleaching or drift correction. Average traces of single mitochondria data were smoothed for graphical representation, but statistical analysis was done on raw data.

## Image analysis

Images were loaded into FIJI/ImageJ (v1.5.3) and background subtracted (radius = 50). Rarely, minor lateral drift was corrected for using the "correct 3D drift" plugin during long-term imaging. Cells which drifted substantially were discarded, but measurements prior to drift were included in the analysis. Each cell was manually traced for region-of-interest (ROI) measurement of fluorescent intensities. Unless otherwise stated, the ratio of HyPer7 emission from 488 nm excitation was divided by the emission from 405 nm excitation and was normalized to same-cell baseline through $H_2O_2$ or photostimulation

experiments. For microdomain $H_2O_2$ diffusion experiments, KillerRed intensity was obtained prior to photostimulation to normalize the same cell HyPer7 response to KillerRed expression. Cells with mitochondrial aggregation or other gross morphological deformities were not included in analysis.

## Mitochondrial morphology

Following background subtraction, images were smoothed with a gaussian filter (sigma = 2). Mitochondria were automatically segmented by a Weka segmentation classifier (v3.3)[93] trained on manually drawn ROIs. TrackMate (v7)[94] was used to report the sum of HyPer7 and KillerRed intensity, area, perimeter, and the major and minor elliptical axis dimensions for each mitochondrion for every frame. The Euclidean distance from the centroid of each mitochondrion per frame to the XY coordinate of spot stimulation was calculated. Data were then analyzed using custom code in R, where the HyPer7 and KillerRed intensities per mitochondrion were normalized to their area to limit artifacts based on Z-drift and protein expression differences. Form factor [perimeter$^2$/(4$\pi$ × area)] was calculated for each mitochondrion in each frame. Rather than use an arbitrary distance cutoff since cellular size and shape vary, mitochondria were classified as proximal and distal based on that mitochondrion's proportion to the maximum distance found from any single mitochondrion to the stimulation spot per cell. Proximal and distal mitochondria were defined as less than 30% and more than 70% of the maximum distance of any mitochondrion to the spot stimulation site, respectively, which corresponded to approximately 5 and 15 $\mu$m. In MEFs, these values were changed to under 20% and over 50%, respectively. For statistical analysis of morphology per frame, only timepoints in which proximal mitochondria were significantly different from both nonstimulated control and distal subpopulation mitochondria were reported. Otherwise, factor-wise ANOVA $p$ values were reported. During some imaging experiments, single frames experienced large technical noise artifacts and were removed from plotting and analysis. Never more than a single frame out of >200 was removed from any given recording.

## *C. elegans* imaging and single mitochondrial linescans

*C. elegans* were maintained at 20 °C on plates containing nematode growth media (NGM) with OP50 bacterial as food[31]. Strains expressing microdomain-targeted HyPer7 with a common IMMT::mCherry background were generated in-house[41]. Staged L4 worms were grown on standard NGM plates and screened for bright expression of HyPer7. Prior to imaging, worms were placed onto a glass slide agar pad containing 10 mM tetramisole hydrochloride and sealed with a No. 1.5 glass coverslip 20 min prior to imaging. Worm hypodermal cell mitochondria were identified by morphology and imaged with 488 and 561 nm laser lines to visualize HyPer7 and mCherry, respectively. HyPer7/mCherry intensities across linescans manually drawn through individual mitochondria were analyzed in ImageJ.

## Statistics

All data were plotted and statistically analyzed in GraphPad Prism (v9) or R (v4.0) and first subjected to a normality test. Single variate analyses were run as either unpaired, two-tailed t-tests or Mann-Whitney U tests. Multivariate analyses were run as either parametric one- or two-way ANOVAs, nonparametric Kruskal-Wallis tests, or repeated measure, mixed-effect two-way ANOVAs with Geisser-Greenhouse correction. All multivariate tests used post-hoc corrections as described in the figure legend text. Outliers were excluded in TMRM response curves to rotenone and antimycin following a 1% ROUT test (9/420 cells). Statistical $p$ values < 0.05 were considered significant with any $p$ values under 0.001 truncated and reported as ***. Relevant statistical data are packaged in the Source Data file.

## Reporting summary

Further information on research design is available in the Nature Portfolio Reporting Summary linked to this article.

## Data availability
All data which supports the findings here can be found in the manuscript and supplementary information. All data generated in this study are provided in the Source Data file. Source data are provided with this paper.

## Code availability
Custom R code used to analyze TrackMate data can be accessed using https://doi.org/10.5281/zenodo.8321362.

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

## Acknowledgements

This work was supported in part by the NIH R01 NS092558, R01 NS115906 (APW). Some strains were provided by the Caenorhabditis Genetics Center (CGC), which is funded by NIH P40 OD010440. JE and GVWJ are supported by NIH R01 AG073121. SAK current affiliation: Harvard University, Department of Neurobiology, Boston, MA 02115, United States of America.

## Author contributions

S.A.K. and A.P.W. conceptualized and oversaw the project. S.A.K. wrote the manuscript. S.A.K., N.A.S., L.D.L.R., J.H., M.A.F., A.Y.W., A.M.E., and J.E. performed experiments. G.V.W.J. contributed resources and funding. All authors edited and approved the final manuscript.

## Competing interests

The authors declare no competing interests.
