## [Peer Review File · Nature Communications]

All-optical spatiotemporal mapping of ROS dynamics across mitochondrial microdomains in situREVIEWER COMMENTS

Reviewer #1 (Remarks to the Author):

The manuscript by Koren et al. suggests a model of how ROS generated in different mitochondrial compartments can spread throughout a cell. The authors also suggest that ROS generation causes mitochondrial hyperfusion and an increase in motility due to local ROS release from mitochondria. These results are interesting and original. The methodology is sound and meets the expected standards. Nevertheless, there are several critical points where the results do not fully support the conclusions and additional evidence is needed.

Does this experimental approach allow to measure the diffusion of ROS between different compartments?

KillerRed stimulation produces superoxide while the Hyper7 detects hydrogen peroxide, and as the authors rightfully mention, the HyPer7 signal is not only reflecting the diffusion of ROS, but also the activity of antioxidant systems in specific compartments. Therefore, it is somewhat risky to conclude that the ROS produced at IMS will move more quickly or efficiently to the matrix than to OMS (Extended Fig.6).

Also, as I understand, this conclusion is mainly based on data presented in Figure 4BDE, showing that half-maximal values after KillerRed activation in IMS will be achieved faster in the matrix than in OMS. However, this seems to be contradictory to results in the same figure showing that diffusion to the matrix is even faster than in IMS, where that ROS is produced. This cannot be explained by higher SOD activity there, which should be lower in IMS?

Are all the observed effects directly related to ROS production, or are they induced/affected by mitochondrial depolarisation?

Authors have not measured mitochondrial membrane potential after KillerRed photostimulation. Stimulation with a 561 nm laser (30% power, 88 μ W, up to 180s) is quite powerful and it cannot be excluded that the laser irradiation could (partially) depolarize mitochondria. Also, the ROS produced by KillerRed could depolarise the mito. If so, then some of the observed effects could be indirectly related to or affected by membrane potential rather than ROS.

Whether the mitochondrial indeed elongate and how?

The results showing the mitochondrial elongation in response to KillerRed photostimulation are not fully convincing. There is a small increase in the area and form factor after stimulation. However, it is not clear from there how many mitochondria were imaged and what statistical test was used. The area before stimulation was higher for stimulated mitochondria, and only very few form factor peak values were higher in the simulated group. More specific analysis is required to conclude that mitochondria elongate. It would also be helpful to understand whether this is really hyperfusion (stimulated mitochondrion fuses with neighboring ones?) or just loss of fission.

The authors also conclude that distal ROS spread is correlated with the rapid trafficking of oxidized mitochondria to distal regions of the cell (Supplemental Movie 3). However, there seems to be no numerical data provided to support this conclusion.

Minor comments:

At the beginning of the Results section, there is a statement that ROS generation is measured in single mitochondria in situ. For clarity, please always indicate which cell lines were used in specific experiments and include this information in figure legends.

Fig.3: There seems to be mislabelling in main text and figure referring to graphs. Row No.255 „...were enough to significantly lower $\Delta\Psi_m$ (Fig. 3C and F) “– should be Fig. 3C and E; and „HyPer7 at any concentration (Fig. 3E). “– should be Fig. 3D instead.

In figure 4C, the authors refer to maximal interpolated steady-state H₂O₂ concentrations in microdomains that were interpolated from exogenous H₂O₂ concentrations. The explanation of how the quantification of H₂O₂ concentration in microdomains was performed remains rather vague and difficult to comprehend. How are the diffusion barriers/differences at antioxidant levels at different compartments taken into account?

Reviewer #2 (Remarks to the Author):

In this study, Koren and colleagues employed a combination of KillerRed-induced superoxide generation and HyPer7-based H₂O₂ monitoring to investigate the handling and spread of H₂O₂ through the mitochondrial network and cells. They thereby observe that mitochondria do release H₂O₂, that induction of superoxide induction in a specific part of the mitochondrial network induced hyperfusion, and that generation of H₂O₂ in specific parts of mitochondria results in distinct responses.

This is a nice, interesting and timely study that in my opinion would raise significant interest of readers interested in H₂O₂ dynamics, signalling and mitochondrial physiology. The analysis of HyPer7 data is exemplary and the data are carefully interpreted (maybe with exception of the H₂O₂ concentration calculation, see below).

The study has in my opinion two main strengths: 1/ the authors use a tool to generate superoxide anions (the proximal ROS produced in mitochondria) and not like in other studies generate directly H₂O₂, and 2/ the authors can induce superoxide anion generation with high spatiotemporal resolution, i.e. in distinct parts of the mitochondrial network within the same cell.

These strengths would have allowed exciting new insights into mitochondrial H₂O₂ signalling and rapid cellular adaptations; however, the study remains remarkably descriptive. In my opinion addressing the following major points experimentally would strengthen the study...

Major points:

1/ Some parts of this study are too descriptive; additional experiments using KO cell lines/siRNAs/inhibitors would strengthen the conclusions of the authors:

1A/ A strength of the experimental setup is the use of KillerRed to produce superoxide anions, the proximal ROS generated in mitochondria (e.g. by the respiratory chain), and not H₂O₂ as DAO does (also pointed out as a major hallmark of their study by the authors in line 71-73). Dismutation of superoxide anions is catalysed by SODs. The enzymatic activity of these enzymes is key to understanding H₂O₂ dynamics originating from the proximal ROS, superoxide anions. The authors should investigate the roles of SOD2 (matrix) and SOD1 (IMS, cytosol) in mitochondrial H₂O₂ dynamics and release.

1B/ Fig. 1I-J: Mitochondria undergo fusion (and obviously also fission) upon stimulation of KillerRed. What happens in cells with impaired fission and fusion machineries. Are the waves of hyperfusion suppressed in these cells? What is the effect on Hyper7 oxidation kinetics and cellular H₂O₂ progression?

2/ Line 267ff: “interpolating H₂O₂ amounts per micro domain”: I am sceptical about the validity of the

approach. HyPer7 does not allow to measure absolute H₂O₂ concentrations and can also not be calibrated due to the reductive half reaction that differs between different compartments...

Moreover, to my understanding the calculation does not take reducing systems into account (that differ quite dramatically between subcompartments). Here, I might be mistaken because the calculation is presented rather short/superficial.

I am also not convinced by the calculated local H₂O₂ concentrations – they are too high with almost 1 μM local H₂O₂. The risk with casually providing numbers is that non-experts will reference them without being aware of the limitations of the calculation.

3/ “Mitochondrial ROS spreads intracellularly through transient hyperfusion and motility” – how long does KillerRed produce superoxide anions after stimulation? Could the observed progression through the mitochondrial network be due to distribution of active superoxide anion-generating KillerRed? How can the minute-long ROS spread be otherwise explained in the presence of efficient antioxidative systems?

Minor points:

1/ The authors tend to “oversell” their study in comparison to previous studies. Enclosed just a few examples:

1A/ Line 52: “kinetics only beginning to be characterized”; There are various experimental studies directly (!) looking at intramitochondrial H₂O₂ dynamics in mammalian cells but also in different fungi using different H₂O₂ sensors

1B/ Line 54: “minimal experimental evidence”; Ref. 12 (Hoehne et al) reports directly on H₂O₂ release on glucose as well as galactose; ref. 28 (Pak et al) also directly observes H₂O₂ release although only if the authors inhibit the powerful reducing systems of the cytosol. Moreover, numerous studies demonstrate effects of H₂O₂ release most of them indirectly although also here some studies provide direct evidence e.g. Sabharwal Biochem J 2013 express PRDX5 in the IMS to attenuate hypoxia-induced ROS signalling effects clearly pointing to a role of mitochondrial H₂O₂ release in cellular signalling.

2/ Line 96: It has not been tested by the developers of HyPer7, Pak et al (Cell Metab 2020) whether HyPer7 can be directly oxidized by superoxide anions. Although this might be unlikely, the authors should mention this.

3/ Line 179: What is the “outer membrane space (OMS)”? The TOM20 targeting signal indicates that HyPer7 was targeted to the cytosolic side of the OMM. This location should also be described like this.

4/ Line 252ff: depletion of the membrane potential likely also affects regeneration of NADPH as the major electron source for antioxidative responses. Thus, I would expect not only ROS production by the respiratory chain but at the same time also a decrease in NADPH replenishing capacity in the matrix.

5/ Line 272: how do the authors precisely define “directionally selective diffusion”?

6/ Line 300ff: The following statement cannot be correct: “ROS made in the OMS, in contrast, diffuses quickest into the matrix before a lag time where it eventually oxidizes the IMS”. The ROS will diffuse through the IMS, it is just not detected by IMS-HyPer7. Besides correcting this, could the authors provide an explanation for this observation.

7/ Line 493: in ref.12 not only glucose but also galactose was used to investigate H₂O₂ handling

REVIEWER COMMENTS

We thank the editor and reviewers for their enthusiasm and thorough critiques as well as for the opportunity to revise our manuscript. We have reproduced the reviewer comments verbatim below in **blue** and provided responses to individual points in black. Articles referenced in our response are listed using PubMed IDs (PMID).

Overall, we have completed experiments suggested by both reviewers to 1) investigate how altering regulators of mitochondrial dynamics through knock-down of DRP1 (to induce hyperfusion by blocking fission) and OPA1 (to induce fragmentation by blocking fusion) alters mitochondrial transient elongation and ROS dynamics seen in our data; 2) knock-down of SOD1/2 levels to determine how cytoplasmic and IMS SOD1 vs matrix SOD2 activity alter ROS diffusion and release, 3) examine the role of mitochondrial depolarization in spatiotemporal ROS photogeneration and diffusion, and 4) test whether photostimulation-mediated mitochondrial depolarization without KillerRed leads to any transient elongation phenotype. To answer these questions, we generated five different cell lines and nearly doubled the amount of data in the manuscript. We are hopeful that the reviewers will recognize our sincere attempt to address their comments and agree that the manuscript is stronger as a result.

Reviewer #1 (Remarks to the Author):

The manuscript by Koren et al. suggests a model of how ROS generated in different mitochondrial compartments can spread throughout a cell. The authors also suggest that ROS generation causes mitochondrial hyperfusion and an increase in motility due to local ROS release from mitochondria. These results are interesting and original. The methodology is sound and meets the expected standards. Nevertheless, there are several critical points where the results do not fully support the conclusions and additional evidence is needed.

We thank the reviewer for their kind summary.

Does this experimental approach allow to measure the diffusion of ROS between different compartments?

KillerRed stimulation produces superoxide while the Hyper7 detects hydrogen peroxide, and as the authors rightfully mention, the HyPer7 signal is not only reflecting the diffusion of ROS, but also the activity of antioxidant systems in specific compartments. Therefore, it is somewhat risky to conclude that the ROS produced at IMS will move more quickly or efficiently to the matrix than to OMS (Extended Fig.6).

We fully agree that it would be risky to make conclusions on the speed of ROS diffusion between mitochondrial compartments based on the HyPer7/KR experiments in this manuscript. We consider the time until half maximal ($T_{1/2}$) value a measure of how rapidly

steady-state is approached given varied antioxidant machinery and not a measure of absolute diffusion speed. As the reviewer points out, we cannot detect ROS that is scavenged by antioxidant systems prior to oxidizing HyPer7 and thus speed is relative to competition with this machinery. **Extended Fig. 6** (now **Extended Fig. 10**) refers to the amount of ROS that has detectable diffusion, rather than speed or efficiency. We have altered text in the manuscript to better reflect this interpretation, see **lines 310-324**.

Also, as I understand, this conclusion is mainly based on data presented in Figure 4BDE, showing that half-maximal values after KillerRed activation in IMS will be achieved faster in the matrix than in OMS. However, this seems to be contradictory to results in the same figure showing that diffusion to the matrix is even faster than in IMS, where that ROS is produced. This cannot be explained by higher SOD activity there, which should be lower in IMS?

We too were surprised by this finding. Biologically speaking, SOD1 and SOD2 both have very fast kinetics and superoxide has limited membrane permeability (PMID: 29669742). Given that, any increase in H₂O₂ can be considered either locally produced or diffused and not due to superoxide diffusing between compartments. We believe the reason for this increased T_{1/2} time in the IMS relative to the matrix is because when ROS enters or is generated at the IMS it quickly diffuses into the matrix. Our data show that ROS generated in the IMS reaches roughly 50% steady-state oxidation of IMS-HyPer7 at roughly the same as Matrix-HyPer7 since the T_{1/2} for Matrix-HyPer7 is the same as IMS-HyPer7 (roughly 3 sec) when ROS is produced in the IMS through IMS-KR. Additionally, our IMS tag includes both the IBM and cristae, so H₂O₂ may diffuse rapidly through the IBM into the matrix instead of diffusing elsewhere in the IMS to oxidize Hyper7 there. We have expanded our explanation of this result in the discussion, see **lines 596-602**.

Are all the observed effects directly related to ROS production, or are they induced/affected by mitochondrial depolarisation?

Authors have not measured mitochondrial membrane potential after KillerRed photostimulation. Stimulation with a 561 nm laser (30% power, 88 μW, up to 180s) is quite powerful and it cannot be excluded that the laser irradiation could (partially) depolarize mitochondria. Also, the ROS produced by KillerRed could depolarise the mito. If so, then some of the observed effects could be indirectly related to or affected by membrane potential rather than ROS.

We agree that 88uW is a moderately high laser power but wish to clarify that the 88 uW stimulation only occurs for 5 sec for at no more than two pulses in a single location per cell over the course of a 20 min recording. The 180 sec illumination as seen in **Fig. 5** is with a much weaker widefield LED. We have now completed experiments testing whether mitochondrial depolarization contributes to our observed phenotypes. To avoid spectral overlap with KR, we used Rhodamine123 to measure the mitochondrial membrane potential (PMID: 6965798). Our photostimulation decreased local Rhodamine123

intensity; though this was not dependent on KR (new **Extended Fig. 2**). To test if this apparent depolarization led to ROS accumulation and transient elongation, we photostimulated cells expressing HyPer7 without KR. The same photostimulation paradigm did not lead to detectable ROS accumulation or marked transient elongation (new **Extended Fig. 3**), see **lines 179-192**.

Whether the mitochondrial indeed elongate and how? The results showing the mitochondrial elongation in response to KillerRed photostimulation are not fully convincing. There is a small increase in the area and form factor after stimulation. However, it is not clear from there how many mitochondria were imaged and what statistical test was used. The area before stimulation was higher for stimulated mitochondria, and only very few form factor peak values were higher in the simulated group.

We thank the reviewer for commenting on these points. Though the change in mitochondrial area is small in absolute number (e.g. from roughly $0.75 \mu\text{m}^2$ to $1 \mu\text{m}^2$ as seen with matrix-KR), this is an increase of 33%--a substantial increase in size of mitochondria. Similarly, form factor describes shape complexity and not necessarily size, so a modest increase from roughly 2.5 to 3 is a 20% increase in complexity, fairly large in already complex shaped mitochondria. We recognize that the averages of mitochondrial area and form factor between conditions and positions in a cell may differ slightly at baseline, though these were not statistically significant between conditions. The transient spikes in area and form factor were enough to drive statistical significance, as noted in the figure. We suspect these minor initial mitochondrial shape differences reflect natural variances in cells, especially given these are not synchronized for cell cycle stage and often imaged across different cell passages. We believe this makes our findings of consistent area and form factor transients in this slightly varied background more meaningful.

Furthermore, we used a conservative statistical test for determining changes in area and form factor by using a two-way ANOVA with Tukey multiple comparison correction. This is considerably more stringent than the field standard of using uncorrected p values given we are imaging area and form factor as continuous variables over nearly 20 minutes of video (200+ frames), instead of a handful of static images at predetermined timepoints. To better aid readers, we have ensured each figure legend explicitly states statistical tests and the number of mitochondria imaged. Across all of our experiments and repeated trials, generally at least 20 mitochondria, on average roughly 50, and up to over 300 are imaged in sum per frame, per condition except for mitochondria directly spot stimulated in **Fig. 5** which is limited to 3-6.

More specific analysis is required to conclude that mitochondria elongate. It would also be helpful to understand whether this is really hyperfusion (stimulated mitochondrion fuses with neighboring ones?) or just loss of fission.

We agree with the reviewer that understanding whether these transient morphological changes are due to increased fusion or decreased fission is an important factor to consider. To better address the question of how (and whether) ROS induces transient mitochondrial fusion vs loss of fission, we developed HEK293T cell lines stably expressing validated shRNAs for either DNML1 coding for DRP1 (to induce hyperfusion) and OPA1 (to induce fragmentation) under puromycin selection. We repeated our experiments using matrix HyPer7 and matrix KillerRed in these cell lines (new **Extended Fig. 4**). Despite puromycin selection leading to a mitochondrial fragmentation phenotype, OPA1 KD further fragmented mitos and slowed HyPer7 reduction following KR stimulation. This slowed reduction matched a complete lack of transient elongation events. On the other hand, DNML1 KD cells with impaired fission had heightened HyPer7 reduction and upregulated elongation events compared to shScrambled-expressing control cells. These results suggested that these transient elongation events were mediated both by fusion and fission machinery but that functional fusion machinery is required for proper acute ROS defense through transient fusion. See **lines 193-213, 532-539**.

The authors also conclude that distal ROS spread is correlated with the rapid trafficking of oxidized mitochondria to distal regions of the cell (Supplemental Movie 3). However, there seems to be no numerical data provided to support this conclusion.

We meant to report this as an available avenue of ROS signaling, not that it was a main driver. We have now further elaborated on this point, see **lines 148-149, 548-553**.

Minor comments:

At the beginning of the Results section, there is a statement that ROS generation is measured in single mitochondria in situ. For clarity, please always indicate which cell lines were used in specific experiments and include this information in figure legends.

This has now been completed.

Fig.3: There seems to be mislabelling in main text and figure referring to graphs. Row No.255 „...were enough to significantly lower $\Delta\Psi_m$ (Fig. 3C and F) “– should be Fig. 3C and E; and „HyPer7 at any concentration (Fig. 3E). “– should be Fig. 3D instead.

We thank the reviewer for pointing this out.

In figure 4C, the authors refer to maximal interpolated steady-state H₂O₂ concentrations in microdomains that were interpolated from exogenous H₂O₂ concentrations. The explanation of how the quantification of H₂O₂ concentration in microdomains was performed remains rather vague and difficult to comprehend. How are the diffusion barriers/differences at antioxidant levels at different compartments taken into account?

We have substantially expanded our explanation of interpolation in the main body of the manuscript, see **lines 309-324**. Briefly, we used the microdomain-specific HyPer7 response curves to increasing H₂O₂ curves as a way to calibrate the level of HyPer7 fluorescence to a given H₂O₂ concentration added to the extracellular media. As discussed in the text, we consider the H₂O₂ response curves of HyPer7 targeted to individual microdomains a description of how much H₂O₂ is available to oxidize HyPer7 after reaching a steady-state following diffusion and reaction with antioxidant machinery. In this way, a stimulus measured by a given microdomain-specific HyPer7 (e.g., matrix-HyPer7) can be used against the matrix-HyPer7 exogenous H₂O₂ curve to interpolate the amount of exogenous H₂O₂ needed to be added to the media to reach the same level of microdomain-specific change in HyPer7, thus taking into account diffusion and antioxidant levels varying between compartments.

Reviewer #2 (Remarks to the Author):

In this study, Koren and colleagues employed a combination of KillerRed-induced superoxide generation and HyPer7-based H₂O₂ monitoring to investigate the handling and spread of H₂O₂ through the mitochondrial network and cells. They thereby observe that mitochondria do release H₂O₂, that induction of superoxide induction in a specific part of the mitochondrial network induced hyperfusion, and that generation of H₂O₂ in specific parts of mitochondria results in distinct responses.

This is a nice, interesting and timely study that in my opinion would raise significant interest of readers interested in H₂O₂ dynamics, signalling and mitochondrial physiology. The analysis of HyPer7 data is exemplary and the data are carefully interpreted (maybe with exception of the H₂O₂ concentration calculation, see below).

The study has in my opinion two main strengths: 1/ the authors use a tool to generate superoxide anions (the proximal ROS produced in mitochondria) and not like in other studies generate directly H₂O₂, and 2/ the authors can induce superoxide anion generation with high spatiotemporal resolution, i.e. in distinct parts of the mitochondrial network within the same cell.

These strengths would have allowed exciting new insights into mitochondrial H₂O₂ signalling and rapid cellular adaptations; however, the study remains remarkably descriptive. In my opinion addressing the following major points experimentally would strengthen the study...

We thank the reviewer for their strong support of our article.

Major points:

1/ Some parts of this study are too descriptive; additional experiments using KO cell lines/siRNAs/inhibitors would strengthen the conclusions of the authors:

1A/ A strength of the experimental setup is the use of KillerRed to produce superoxide anions, the proximal ROS generated in mitochondria (e.g. by the respiratory chain), and not H₂O₂ as DAO does (also pointed out as a major hallmark of their study by the authors in line 71-73). Dismutation of superoxide anions is catalysed by SODs. The enzymatic activity of these enzymes is key to understanding H₂O₂ dynamics originating from the proximal ROS, superoxide anions. The authors should investigate the roles of SOD2 (matrix) and SOD1 (IMS, cytosol) in mitochondrial H₂O₂ dynamics and release.

We have now completed these analyses (**Extended Fig. 8**). Knocking-down SOD1 and SOD2 had strong effects on the rise time of HyPer7 among each microdomain and with either matrix and IMS KR. Notably, this suggests that there is differences in diffusion due to alterations in SOD levels, though this may not be directly due to diminished superoxide dismutation. Instead, we suspect these differences may be due to cell and mitochondrial dysfunction following prolonged knock-down of SODs which are known to be toxic, see **lines 383-398**.

1B/ Fig. 1I-J: Mitochondria undergo fusion (and obviously also fission) upon stimulation of KillerRed. What happens in cells with impaired fission and fusion machineries. Are the waves of hyperfusion suppressed in these cells? What is the effect on Hyper7 oxidation kinetics and cellular H₂O₂ progression?

This was similarly requested by Reviewer 1. Data on cells with impaired fusion (OPA1 KD) or fission (DNML1 encoding DRP1 protein) is now present in **Extended Fig. 4**. Briefly, inhibiting fusion machinery slowed HyPer7 reduction and blocked transient elongation, whereas inhibition fission machinery hastened HyPer7 reduction and facilitated transient elongation, supporting our original claim that transient elongation can act as an acute way to clear ROS, see **lines 193-213, 532-539**.

2/ Line 267ff: “interpolating H₂O₂ amounts per micro domain”: I am sceptical about the validity of the approach. HyPer7 does not allow to measure absolute H₂O₂

concentrations and can also not be calibrated due to the reductive half reaction that differs between different compartments... Moreover, to my understanding the calculation does not take reducing systems into account (that differ quite dramatically between subcompartments). Here, I might be mistaken because the calculation is presented rather short/superficial.

As similarly requested by Reviewer 1, we have expanded our explanation of microdomain-specific interpolation of HyPer7 responses in the main body of the text, **lines 309-324**.

I am also not convinced by the calculated local H₂O₂ concentrations – they are too high with almost 1 μM local H₂O₂. The risk with casually providing numbers is that non-experts will reference them without being aware of the limitations of the calculation.

We fully agree with this point. We did not mean imply calculations of estimated local H₂O₂ concentrations, but rather the estimated final concentration of exogenously applied H₂O₂ into the cellular media needed to approximate the same HyPer7 fluorescence change in that microdomain. As such, we have changed this phrasing on relevant figures and mentions to “exogenous H₂O₂ equivalent” which we believe more accurately describes the metric. We further explain this difference in the results section of the manuscript and add “exogenous equivalents” where appropriate, notably see **lines 324-327**.

3/ “Mitochondrial ROS spreads intracellularly through transient hyperfusion and motility” – how long does KillerRed produce superoxide anions after stimulation? Could the observed progression through the mitochondrial network be due to distribution of active superoxide anion-generating KillerRed? How can the minute-long ROS spread be otherwise explained in the presence of efficient antioxidative systems?

As a photosensitizer, KillerRed does not produce superoxide anions after photostimulation has ended since it no longer has enough energy to continue generating triplet state electrons to transfer onto oxygen (PMIDs: 16369538, 31841676). We believe the prolonged spread of ROS is not necessarily unreacted H₂O₂, but instead oxidized or otherwise damaged cargo shuffling between mitochondria which undergoes slower clearance.

Minor points:

1/ The authors tend to “oversell” their study in comparison to previous studies. Enclosed just a few examples:

1A/ Line 52: “kinetics only beginning to be characterized”; There are various experimental studies directly (!) looking at intramitochondrial H₂O₂ dynamics in

mammalian cells but also in different fungi using different H₂O₂ sensors

1B/ Line 54: “minimal experimental evidence”; Ref. 12 (Hoehne et al) reports directly on H₂O₂ release on glucose as well as galactose; ref. 28 (Pak et al) also directly observes H₂O₂ release although only if the authors inhibit the powerful reducing systems of the cytosol. Moreover, numerous studies demonstrate effects of H₂O₂ release most of them indirectly although also here some studies provide direct evidence e.g. Sabharwal Biochem J 2013 express PRDX5 in the IMS to attenuate hypoxia-induced ROS signalling effects clearly pointing to a role of mitochondrial H₂O₂ release in cellular signalling.

We thank the reviewer for these suggestions. We have included all suggestions raised here.

2/ Line 96: It has not been tested by the developers of HyPer7, Pak et al (Cell Metab 2020) whether HyPer7 can be directly oxidized by superoxide anions. Although this might be unlikely, the authors should mention this.

Pak, et al have tested this by incubating HyPer7 with xanthine oxidase (“XOX”), a superoxide anion generator. HyPer7 appeared to have minimal reactivity when XOX was co-incubated with catalase to convert any H₂O₂ generated by spontaneous dismutation, thereby rendering only superoxide available to potentially react with HyPer7. This point was elaborated on in **lines 479-483**.

3/ Line 179: What is the “outer membrane space (OMS)”? The TOM20 targeting signal indicates that HyPer7 was targeted to the cytosolic side of the OMM. This location should also be described like this.

This has now been fixed throughout the manuscript and in the figures.

4/ Line 252ff: depletion of the membrane potential likely also affects regeneration of NADPH as the major electron source for antioxidative responses. Thus, I would expect not only ROS production by the respiratory chain but at the same time also a decrease in NADPH replenishing capacity in the matrix.

We have included this point, **lines 290-291**.

5/ Line272: how do the authors precisely define “directionally selective diffusion”?

We define directionally selective diffusion as diffusion that is unequally favoring diffusion into one direction (or in this case, compartment) over another. We have now elaborated on our definition of directionally selective diffusion at its first mention, **lines 83-84**.

6/ Line 300ff: The following statement cannot be correct: “ROS made in the OMS, in contrast, diffuses quickest into the matrix before a lag time where it eventually oxidizes the IMS”. The ROS will diffuse through the IMS, it is just not detected by IMS-HyPer7. Besides correcting this, could the authors provide an explanation for this observation.

We have now rephrased the statement for clarity, see **lines 596-602**. One potential caveat to this observation is that HyPer7 is more localized within cristae (generally, as they're brighter) over the thinner inner boundary membrane (IBM) part of the IMS. If H₂O₂ diffuses from the OMM into the IBM, we speculate it would have considerably less time and fewer available HyPer7 molecules to oxidize before diffusing through the IMM through this route compared to the OMM to IBM to cristae to matrix route, where it needs to diffuse farther and will likely interact with more HyPer7 molecules along the way.

7/ Line 493: in ref.12 not only glucose but also galactose was used to investigate H₂O₂ handling

This has been corrected.

REVIEWERS' COMMENTS

Reviewer #1 (Remarks to the Author):

The authors have adequately addressed the criticisms, and the revised edition shows significant improvements.

Reviewer #2 (Remarks to the Author):

this is the revised version of a manuscript that I previously reviewed. The authors have adressed all my points mostly in experiments and have added a significant body of new data. I think the manuscript thereby has strongly improved and unclear points have been clarified. I support publication.

REVIEWER COMMENTS

We thank the editor and reviewers for their enthusiasm and thorough critiques as well as for the opportunity to revise our manuscript. We have reproduced the reviewer comments verbatim below in **blue** and provided responses to individual points in black.

Reviewer #1 (Remarks to the Author):

The authors have adequately addressed the criticisms, and the revised edition shows significant improvements.

We thank the reviewer for the supportive comment.

Reviewer #2 (Remarks to the Author):

this is the revised version of a manuscript that I previously reviewed. The authors have adressed all my points mostly in experiments and have added a significant body of new data. I think the manuscript thereby has strongly improved and unclear points have been clarified. I support publication

We thank the reviewer for the supportive comment.